# MLLM-4D: Towards Visual-based Spatial-Temporal Intelligence

**Xingyilang Yin** [1,2,*] **Chengzhengxu Li** [3,*] **Jiahao Chang** [4] **Chi-Man Pun** [1,†] **Xiaodong Cun** [2,†]

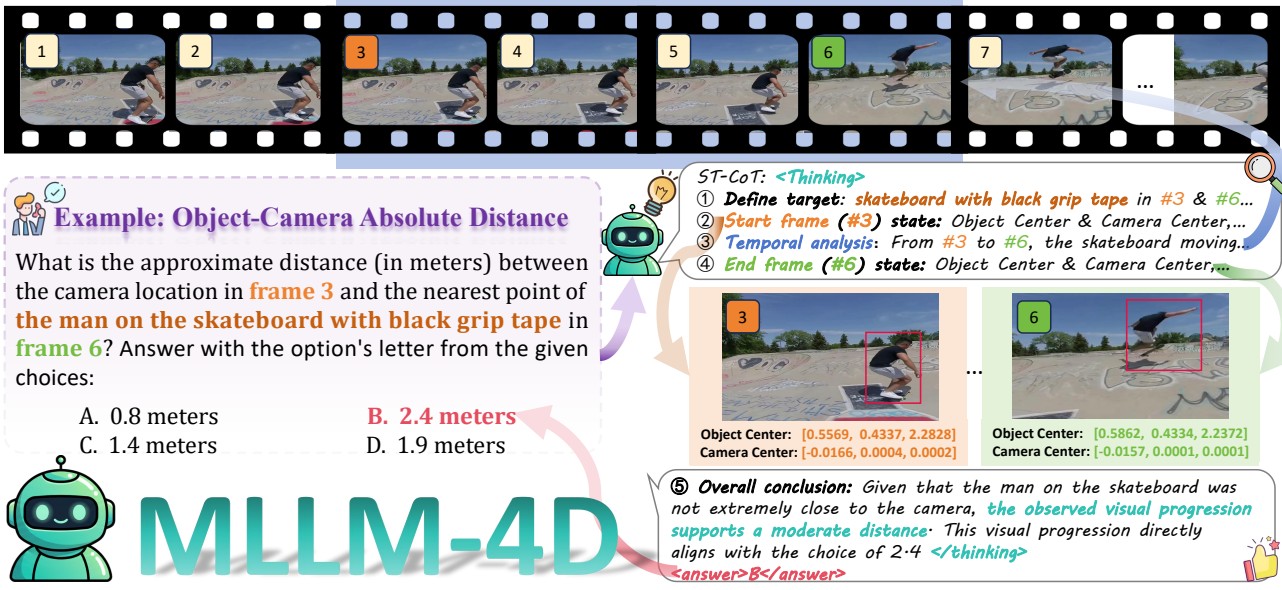

*Figure 1.* We propose *MLLM-4D*, a method that advances MLLMs for the visual-based spatial-temporal intelligence. MLLM-4D is capable of understanding and reasoning about the evolution of 3D space over time from only 2D video input.

## Abstract

Humans are born with vision-based 4D spatial-temporal intelligence, which enables us to perceive and reason about the evolution of 3D space over time from purely visual inputs. Despite its importance, this capability remains a significant bottleneck for current multimodal large language models (MLLMs). To tackle this challenge, we introduce *MLLM-4D*, a comprehensive framework designed to bridge the gaps in *training data curation* and *model post-training* for spatiotemporal *understanding* and *reasoning*. On the data front, we develop a cost-efficient data curation pipeline that repurposes existing stereo video datasets into high-quality 4D spatiotemporal instructional data. This results in the *MLLM4D-2M* and *MLLM4D-R1-30k* datasets for Supervised Fine-Tuning (SFT) and Reinforcement Fine-Tuning (RFT), alongside *MLLM4D-Bench* for comprehensive evaluation. Regarding model training, our post-training strategy establishes a foundational 4D understanding via SFT and further catalyzes 4D reasoning capabilities by employing Group Relative Policy Optimization (GRPO) with specialized *Spatiotemporal Chain of Thought (ST-CoT)* prompting and *Spatiotemporal reward functions (ST-reward)* without involving the modification of architecture. Extensive experiments demonstrate that MLLM-4D achieves state-of-the-art spatial-temporal understanding and reasoning capabilities from purely 2D RGB inputs. Project page: https://github.com/GVCLab/MLLM-4D.

*Equal contribution [1]University of Macau [2]GVC Lab, Great Bay University [3]Xi'an Jiaotong University [4]The Chinese University of Hong Kong, Shenzhen. Correspondence to: Chi-Man Pun <cmpun@umac.mo>, Xiaodong Cun <cun@gbu.edu.cn>.

*Proceedings of the 43rd International Conference on Machine Learning*, Seoul, South Korea. PMLR 306, 2026. Copyright 2026 by the author(s).

## 1. Introduction

Humans possess an innate 4D spatiotemporal intelligence that extends beyond the perception of static 3D geometry to integrate time as an intrinsic cognitive dimension. This abil-

ity to reason about the world in 4D (3D space + time) allows us to navigate and act effectively within constantly changing environments using purely visual inputs. Such ability is critical to interactive AI systems, including VR/AR and embodied agents, where navigating dynamic scenes requires a continuous understanding of evolving spatial relationships. While Multimodal Large Language Models (MLLMs) have demonstrated remarkable general intelligence (Hurst et al., 2024; Comanici et al., 2025) in image (Liu et al., 2023; Li et al., 2024), video (Bai et al., 2025b;a), and audio (Chu et al., 2024; Xu et al., 2025a), their abilities for this spatiotemporal intelligence remain largely underexplored.

Recent works mainly focus on spatial reasoning within static scenes (Yang et al., 2025; Dihan et al., 2025; Yang et al., 2026a; Azuma et al., 2022; Ma et al., 2022; Zhang et al., 2025) and struggle to understand and reason about evolving relationships within 4D space, as illustrated in Fig. 1. For dynamic scenarios, manual annotations can only collect small benchmark-size datasets (Zhou et al., 2025b; Li et al., 2025) and are challenging to scale for current MLLM training. On the other hand, current methods enhance 3D spatial intelligence of MLLMs using additional spatial encoders (Zheng et al., 2025; Fan et al., 2025; Huang et al., 2024; Deng et al., 2025; Zhu et al., 2025; Liu et al., 2025). However, these 3D expertise MLLMs often fail in dynamic reasoning tasks, as their learned knowledge is constrained to static environments with immobile objects.

We thus introduce *MLLM-4D*, a novel and comprehensive framework for boosting MLLM capabilities for visual-based spatiotemporal intelligence by addressing two critical bottlenecks introduced above: *(i) the scarcity of large-scale and high-quality 4D instructional data.* We develop an automated data engine that repurposes existing stereo video datasets (Shao et al., 2024) into high-quality 4D spatiotemporal instructional data. Our pipeline integrates several advanced vision primitives to decompose these scenes into per-frame camera poses, object-level 3D points and corresponding semantic descriptions, capturing the rich 4D evolution of the entire scene. By applying rigorous physics-based spatiotemporal computations to this metadata, we generate a large-scale *MLLM4D-2M* dataset for Supervised Fine-Tuning (SFT), *MLLM4D-R1-30k* dataset for Reinforcement Fine-Tuning (RFT), and *MLLM4D-Bench* for comprehensive evaluation, respectively. *(ii) the lack of specialized and scalable 4D-aware MLLMs.* Contrary to prior works that rely on auxiliary spatial encoders, we demonstrate that standard MLLM architectures can achieve robust spatiotemporal reasoning when scaled with high-quality 4D data. Our framework utilizes a hierarchical post-training approach to associate pixel-level video observations with 4D physical reasoning. In the first stage, SFT establishes a foundational 4D understanding, ensuring the model can correctly identify spatial-temporal anchors. In the second stage, we catalyze

advanced 4D reasoning capabilities through Group Relative Policy Optimization (GRPO) (Liu et al., 2024a). We introduce a specialized five-step *Spatiotemporal Chain of Thought (ST-CoT)* prompting that forces the model to act as a visual physics engine, focusing on temporal anchoring, 3D state parsing, and physical motion. We move beyond standard accuracy reward and format reward by introducing *Spatiotemporal reward (ST-reward)* functions. This reward serves as a physical regularizer, penalizing the model for hallucinated motion that contradicts the actual spatiotemporal evolution of the scene. Extensive experiments demonstrate that MLLM-4D achieves state-of-the-art 4D spatiotemporal understanding and reasoning performance.

Our main contributions are summarized as follows:

- We introduce *MLLM-4D*, a novel and comprehensive framework that significantly enhances the spatial-temporal intelligence of MLLMs, demonstrating strong 4D understanding and reasoning capabilities without requiring architectural modifications.

- We develop an *automated data curation pipeline* to generate high-quality 4D spatiotemporal instructional data by repurposing the existing stereoscopic video datasets. Leveraging this pipeline, we propose the *MLLM4D-2M* and *MLLM4D-R1-30k* datasets for SFT and RFT, alongside *MLLM4D-Bench* for comprehensive evaluation.

- In our training framework, we propose specialized *ST-CoT* prompting strategies and physics-grounded *ST-reward*. These are integrated into GRPO to systematically improve the model's capacity for verifiable 4D spatiotemporal reasoning in dynamic scenes.

- Extensive experiments demonstrate that our MLLM-4D achieves state-of-the-art 4D understanding and reasoning performance with only RGB video input.

## 2. Related Works

**Multimodal Large Language Models .** Multimodal Large Language Models (MLLMs) (Liu et al., 2023; Li et al., 2023; Zhang et al., 2024; Hurst et al., 2024; Tong et al., 2024; Wang et al., 2025e; Zhou et al., 2025a; Comanici et al., 2025) have achieved remarkable success across diverse 2D visual tasks. Recent advancements, such as Qwen3-VL (Bai et al., 2025a) achieve strong visual modeling across images and video by integrating interleaved-MRoPE, multi-level visual features, and text-based time alignment. Despite these strides, even the state-of-the-art MLLMs struggle to interpret the complex underlying 4D scene from video. Our proposed MLLM-4D is designed to bridge this gap, enhancing the foundational capabilities of existing MLLMs with 4D spatiotemporal understanding and reasoning.

**MLLMs for Spatial Intelligence.** Recent advances (Ma et al., 2025; Shen et al., 2025; Xu et al., 2025b; Ouyang et al., 2025; Yang et al., 2026b) have sparked interest in extending MLLMs to encompass 3D spatial understanding and reasoning. While some methods rely on auxiliary 3D geometric input (Huang et al., 2024; Deng et al., 2025) or 2.5D depth information (Zhu et al., 2025; Liu et al., 2025), recent studies including VG-LLM (Zheng et al., 2025), Spatial-MLLM (Wu et al., 2025a), and VLM-3R (Fan et al., 2025) attempt to perceive the 3D world directly from video by leveraging 3D reconstruction priors (Wang et al., 2025a;b). However, these 3D expertise MLLMs remain largely constrained to static scenes with immobile objects and struggle to learn the evolving relationships within a 4D spatiotemporal manifold. In contrast, our MLLM-4D establishes a foundational 4D understanding and reasoning capabilities with our dataset and designed training recipes.

**Visual-based 4D Spatial-Temporal Intelligence.** Visual-based spatial-temporal intelligence focuses on enabling video MLLMs to understand and reason about 4D spatiotemporal relationships directly from visual input. Previous works mainly focus on improving the spatial intelligence of MLLMs through 3D QA datasets (Azuma et al., 2022; Ma et al., 2022; Zhang et al., 2025) on static 3D spatial reasoning benchmarks (Yang et al., 2025; Dihan et al., 2025; Yang et al., 2026a; Jia et al., 2025). More recently, efforts (Zhou et al., 2025b; Li et al., 2025) such as VLM4D (Zhou et al., 2025b) have begun to evaluate the spatiotemporal reasoning capabilities of MLLMs. However, these benchmarks are limited to a few thousand QA pairs and rely on manual annotation, which lacks the scalability required for MLLM fine-tuning. To address the data constraints, we propose an automated data curation pipeline to generate large-scale training and evaluation datasets, establishing a high-quality foundation for 4D spatiotemporal intelligence learning.

## 3. Scalable Spatial-Temporal Data Curation

We utilize the question-answer (QA) pair as the fundamental training type to enhance the spatial-temporal understanding and reasoning of MLLM. However, current 4D instructional datasets (Zhou et al., 2025b; Li et al., 2025) are typically limited to benchmark-scale samples (e.g, about 2k in VLM4D), as they rely on manual annotations with unstructured types, which are not scalable for training MLLMs.

Thus, we first define the included scope of the spatial-temporal intelligence from the decoupling aspect of camera and object movement. As shown in Fig. 2, which includes three distinct categories:

*(i) Independent Object Motion* has one type of question, which aims to ask about the objects' absolute distance changes over time (frames).

*(ii) Camera Ego-Motion* assesses the camera's displacement and orientation changes through two metrics: *camera ab-*

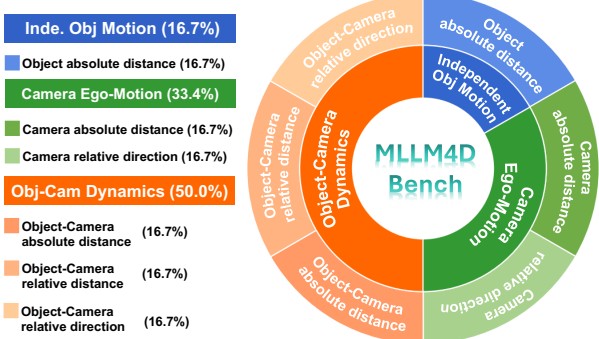

*Figure 2.* The components of our MLLM4D-Bench.

*solute distance*, which measures the physical span of the camera's movement among frames, and *camera relative direction*, which tracks the angular trajectory of the camera over temporal dimension (frames).

*(iii) Object-Camera Dynamics.* This category characterizes the intricate spatial interplay between the camera and moving objects. It encompasses three core dimensions: the absolute distance between the two, their relative distance change, and their relative angular orientation.

To automatically obtain the scalable and accurate question-answer pairs, we predict the spatial-temporal metadata from the video and obtain the answer via physical laws. This metadata contains the per-frame camera poses, object-level 3D point clouds, and fine-grained semantic descriptions, which constitutes a comprehensive 4D representation of dynamic scenes. However, directly processing monocular videos through 4D tracking models (Xiao et al., 2025; Badki et al., 2025) often suffers from depth ambiguity and accuracy issues. Instead, we introduce an automated pipeline that extracts precise 4D spatiotemporal information from existing stereoscopic video datasets (Jin et al., 2025). After obtaining the metadata, we utilize the physical-based formulations to solve for exact spatiotemporal relationships and obtain the thinking process via the MLLM-based CoT generation pipeline, alongside the MLLM4D-Bench for evaluation. Below, we give the details of each part.

**4D Spatial-Temporal Metadata from Stereo Videos.** As illustrated in Fig. 3, giving the stereoscopic video from Stereo4D (Jin et al., 2025), we obtain the $K$ left-rectified video frames $\{I_i\}_{i=1}^K$; processed camera pose (Schonberger & Frahm, 2016) $\{C_i\}_{i=1}^K$, where $C_i = [R_i|t_i]$ consists of a $3 \times 3$ rotation matrix $R_i$ and a $3 \times 1$ translation vector $t_i$; per-frame metric 3D points $\{P_i\}_{i=1}^K$ from metric stereo depth (Wang et al., 2024) and point tracking (Doersch et al., 2024) to obtain the stereo metadata. This is followed by a robust filtering stage to filter out the low-quality estimations.

After that, to obtain the instance-level semantic annotation from a question-answer pair, we employ a video MLLM, *i.e.* Gemini-2.5-flash (Comanici et al., 2025), to identify all moving entities and extract their corresponding noun

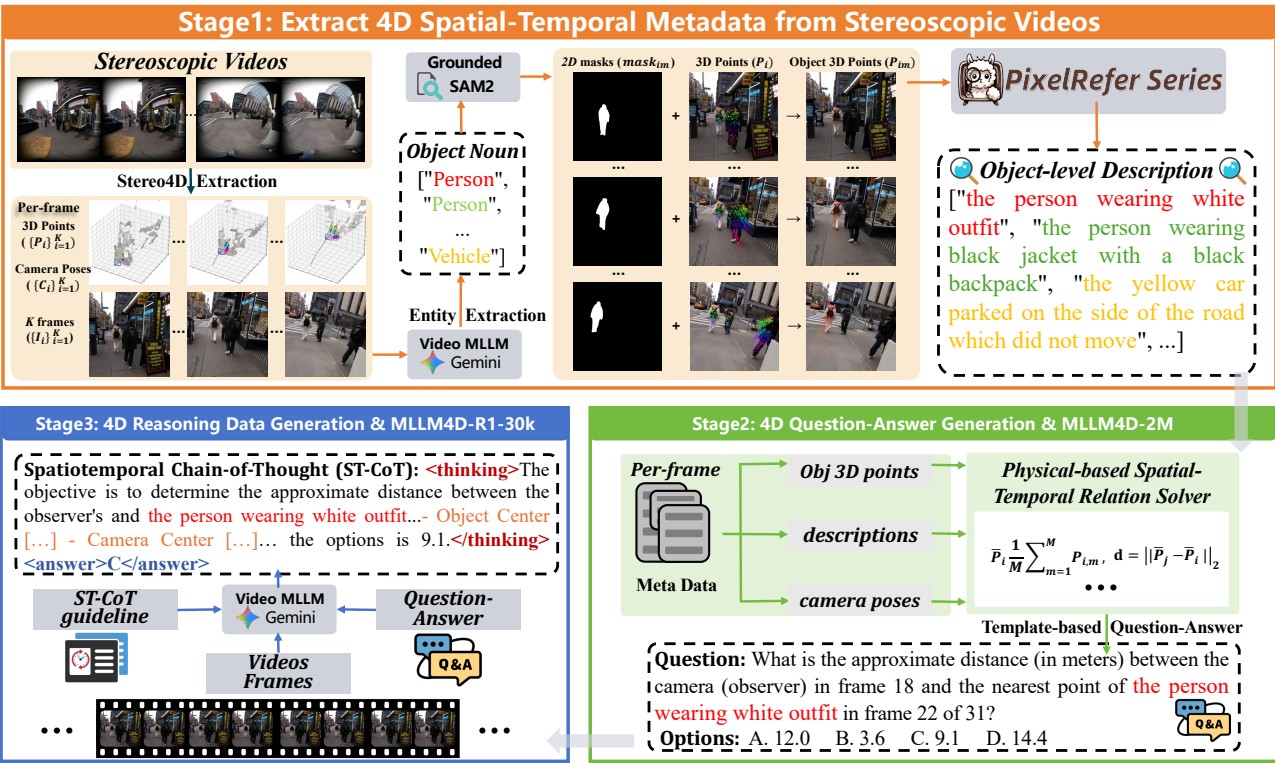

Figure 3. **Our scalable curation pipeline for instructional spatiotemporal data.** Our automated pipeline leverages several advanced vision techniques to extract 4D spatiotemporal information from stereoscopic videos, including per-frame camera poses, object-level 3D point clouds, and semantic descriptions. These data are then processed through a physics-based spatiotemporal relation solver to generate 4D QA pairs, and our specialized ST-CoT prompting strategy synthesizes the corresponding reasoning trajectories.

categories (see Appendix 7.2 for details). We then utilize GroundedSAM2 (Ren et al., 2024; Ravi et al., 2025; Liu et al., 2024b) for instance segmentation and tracking, yielding temporally consistent 2D masks across the sequence. Finally, the scene-level 4D points $\{P_i\}_{i=1}^{K}$ are projected onto these 2D masks $\{mask_{im}\}_{i=1,\ldots,K;m=1,\ldots,M}$ to isolate per-frame, object-level 4D points for M distinct objects, denoted as $\{P_{im}\}_{i=1,\ldots,K;m=1,\ldots,M}$. To enrich these representations, the video frames and their associated 2D masks are fed into a region-level MLLM, PixelRefer (Yuan et al., 2025), to generate fine-grained semantic descriptions $\{T_m\}_{m=1}^{M}$ for each object. The camera pose, fine-grained description, and the instance-level 3D points build the general form of the metadata for further spatial-temporal relationship solver via physical laws.

**Physical-based Spatial-Temporal Relationship Solver.** To generate high-fidelity QA pairs, we derive ground-truth results through rigorous physics-based spatial-temporal computation. For instance, to determine the Camera Relative Direction between any two frames $i$ and $j$: we first compute the world-space displacement $\Delta t = t_j - t_i$. To determine the movement relative to the camera's perspective at frame $i$, we project this vector into the local camera coordinate system: $d_{cam\_rel\_dir} = R_i^T \cdot \Delta t$. The resulting vector $d_{cam\_rel\_dir} = [dx, dy, dz]$ is interpreted according to stan-

dard camera conventions, where $+X, +Y, +Z$ correspond to the right, down, and forward directions, respectively. Please refer to Appendix 7.2 for more details about other spatiotemporal relation solver implementations.

**MLLM4D-2M Dataset.** After calculating the ground-truth values, we apply templates to formulate the final QA pairs. Please refer to the Appendix 7.2 for more details about templates and data filtering details. After filtering, we retain $2M$ high-quality QA pairs across approximately $100k$ videos, forming the large-scale *MLLM4D-2M* dataset for supervised fine-tuning (SFT).

**MLLM4D-Bench.** We report the composition and evaluation settings of our MLLM4D-Bench in Fig. 2, which comprise 6k questions organized into six specialized subtasks. Our benchmark distinguishes itself from existing evaluation suites in several key dimensions: (1) Dynamic Scene Complexity: Unlike current 3D spatial benchmarks (Yang et al., 2025; 2026a; Jia et al., 2025) which focus on reasoning within static environments, MLLM4D-Bench evaluates dynamic scenes involving both moving camera perspectives and multiple moving objects; (2) Structured 4D Motion Categorization: Compared to existing 4D benchmarks (Zhou et al., 2025b; Li et al., 2025), we provide a more rigorous decomposition of 4D motion into three logical cate-

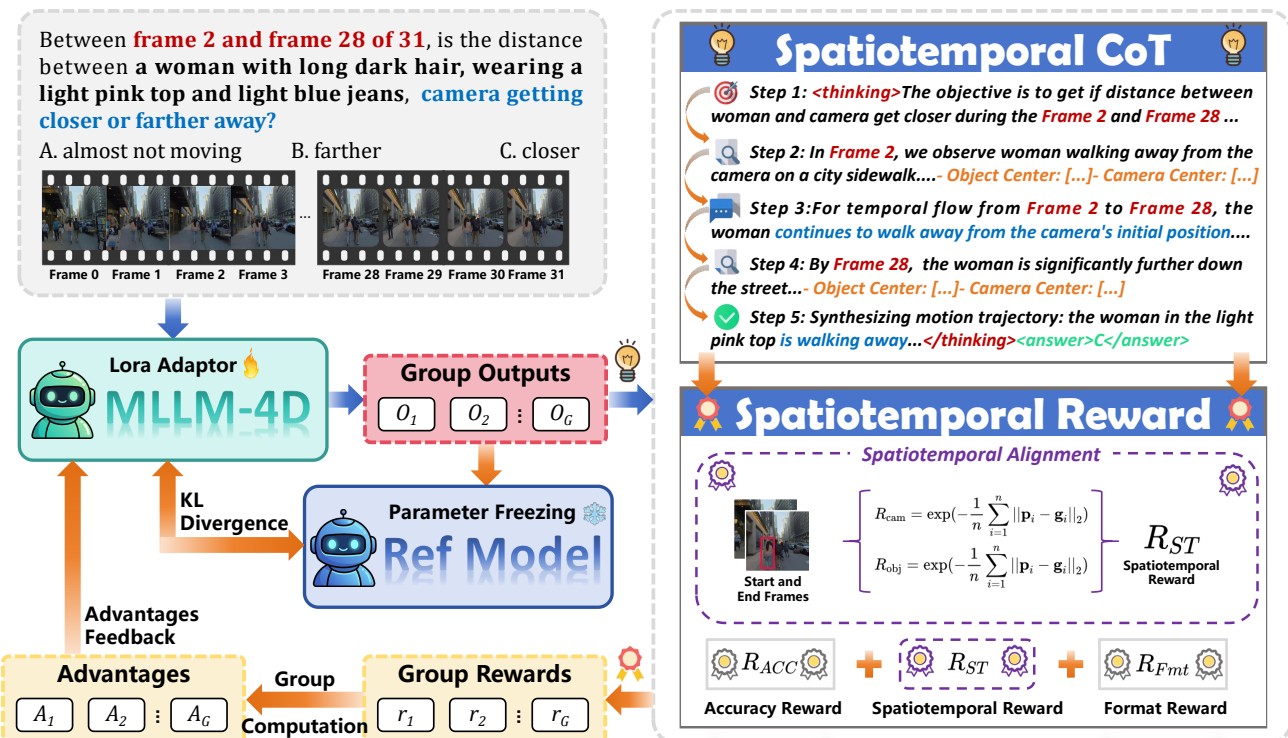

*Figure 4.* **Our RFT pipeline.** Given the input video and question, the MLLM-4D model generates multiple rollouts using the ST-CoT reasoning format. Within each group, relative advantages are computed based on accuracy reward, format reward and ST-reward. The model parameters are then updated via the GRPO objective, which incorporates a KL penalty relative to the frozen reference model.

gories: Independent Object Motion, Camera Ego-Motion, and Object-Camera Dynamics; (3) Fine-Grained Temporal Evaluation: Our benchmark features fine-grained, frame-wise temporal tagging. This enables the precise evaluation of 4D spatiotemporal reasoning between any two arbitrary frames in a video, which is absent in previous benchmark.

**4D Reasoning Data Generation.** We synthesize detailed thinking processes to align the model for 4D spatiotemporal reasoning. By prompting Gemini-2.5-Pro with video frames, corresponding QA pairs, 4D physical values, and spatiotemporal CoT guideline, we generate specialized spatiotemporal CoT data (See Sec. 4.2, Fig. 11 and Appendix 7.3 for details). This process yields 7k cold-start samples and the *MLLM4D-R1-30k*, which contains 30k QA pairs with significant 4D motion and ground-truth solutions designed for large-scale Reinforcement Fine-Tuning.

## 4. MLLM-4D Framework

Previous works involve additional spatial encoders (Zheng et al., 2025; Fan et al., 2025; Wu et al., 2025a) to boost spatial understanding. Notably, we find that using a *pure 2D visual encoder and retaining the standard MLLM architecture already achieves* sota performance when supported by our scalable, high-quality spatial-temporal datasets and optimized post-training framework. We propose a two-stage post-training framework for both understanding and reasoning. In the first stage, we conduct SFT on MLLM4D-2M to

establish foundational 4D understanding (Sec. 4.1). To further enhance 4D reasoning, we utilize a cold start phase to align the model's output with our specialized *Spatiotemporal Chain of Thought (ST-CoT)* (Sec. 4.2). This is followed by the second stage in which we employ GRPO, leveraging the ST-CoT prompting and our *spatiotemporal reward (ST-reward)* functions on MLLM4D-R1-30k (Sec. 4.3).

### 4.1. Supervised Fine-Tuning for 4D Understanding

Leveraging the proposed MLLM4D-2M dataset, we first perform supervised fine-tuning (SFT) to establish foundational 4D spatiotemporal comprehension. To ensure efficient adaptation while preserving the model's pre-trained multimodal knowledge, we utilize the Low-Rank Adaptation (LoRA) (Hu et al., 2022) technique, where trainable rank-decomposition matrices are injected into the linear layers of Transformers. During this stage, we employ the standard cross-entropy loss $\mathcal{L}_{\text{ce}}$ between the model-generated answer sequences and the ground-truth annotations:

$$\mathcal{L}_{\text{ce}} = -\sum_j \log P(o^{(j)} \mid o^{(1:j-1)}, q, \{I_i\}_{i=1}^{N_k}), \quad (1)$$

where $\{I_i\}_{i=1}^{N_k}$ denotes the sequence of $N_k$ input video frames, $q$ represents the concatenated system prompt and scenario-specific question, $o^{(i)}$ denotes the $i$-th token in the target reasoning-answer trajectory, and $o^{(1:i-1)}$ denotes the preceding tokens. This foundational phase ensures the model internalizes the prerequisite spatial-temporal alignment necessary for subsequent high-level reasoning.

## 4.2. Cold-Start Alignment for 4D Reasoning

As shown in Fig. 4, to transition from basic perception to complex 4D analysis, we introduce the *Spatiotemporal Chain-of-Thought (ST-CoT)*, a reasoning paradigm designed to **associate 2D pixel-level observations with 4D world-state representations** (See Fig. 11 for detail prompting settings). Unlike conventional CoT approaches (Wei et al., 2022) that primarily navigate linguistic logic, ST-CoT compels the model to operate as a 3D perception engine, grounding its internal reasoning in visual physics and motion dynamics over time. We utilize this framework to curate a specialized cold-start dataset (see Appendix 7.3 for details). By performing cold-start alignment on this structured data using the same LoRA parameter and loss objective $\mathcal{L}_{ce}$ as the SFT stage, we establish a policy initialization that serves as a stable foundation for subsequent reinforcement learning. The ST-CoT guideline follows the five-step logical flow:

*Step 1: Objective Alignment and Temporal Anchoring:* The reasoning process begins by explicitly defining the spatiotemporal objective $O_{obj}$. The model identifies the query's intent and locks onto critical temporal boundaries (the *start frame* $t_{start}$ and *end frame* $t_{end}$), which prevent computational drift during video processing.

*Step 2: Start frame 3D State Parsing and Anchoring:* At the start frame $t_{start}$, the model performs a joint visual and geometric analysis to output the initial spatial state $\mathcal{S}_{t_{start}}$. By anchoring the camera center, object center, and scene descriptions, the model establishes a quantitative baseline for all subsequent motion estimations.

*Step 3: Temporal Progression and Visual Cue Collection:* Rather than relying on implicit visual flow, the model rationalizes the physical shift between boundaries. It analyzes visual transformations, such as scale expansion and perspective distortion, to infer underlying geometric deltas $\mathcal{T}_{motion}$, creating a causal bridge between visual observation and 4D physical motion.

*Step 4: End frame 3D State Verification:* Upon reaching $t_{end}$, the model generates the terminal spatial state $\mathcal{S}_{t_{end}}$ and a corresponding visual summary. This serves as a consistency check; by comparing $\mathcal{S}_{t_e}$ with $\mathcal{S}_{t_s}$, the model validates the continuity of the spatiotemporal trajectory and minimizes temporal hallucinations.

*Step 5: Evidence-Based Synthesis and Probabilistic Inference:* Finally, the model synthesizes the accumulated visual evidence into a comprehensive trajectory. We formally define the ST-CoT as the coherent sequence $\mathcal{T}_{ST} = \{O_{obj}, \mathcal{S}_s, \mathcal{T}_{motion}, \mathcal{S}_e\}$, aggregating the objective $O_{obj}$, spatial anchors ($\mathcal{S}_s$ & $\mathcal{S}_e$), and temporal motion ($\mathcal{T}_{motion}$). The final answer $\hat{a}$ is derived through a maximum a posteriori estimation, conditioned strictly on this reconstructed 4D trajectory and the raw video input $\mathcal{V}$:

$$\hat{a} = \arg\max_{a \in \mathcal{A}} P(a \mid \underbrace{O_{obj}, \mathcal{S}_s, \mathcal{T}_{motion}, \mathcal{S}_e}_{\mathcal{T}_{ST}}, \mathcal{V}; \theta). \quad (2)$$

This formulation ensures that the final output is not a hallucinated guess but a logical derivative of the explicit 4D progression. By forcing the model to justify its answer through the established trajectory $\mathcal{T}_{ST}$, we enhance the interpretability and capability of the 4D reasoning task.

## 4.3. Reinforcement Fine-Tuning for 4D Reasoning

Building upon the policy initialization from the cold-start phase, we employ GRPO (Shao et al., 2024; Liu et al., 2024a) to further enhance the model's reasoning capabilities, leveraging the proposed MLLM4D-R1-30k dataset. Unlike traditional Actor-Critic frameworks, GRPO eliminates the need for a separate value function by utilizing the relative rewards within a sampled group. For each input $q$, we sample a group of $G$ outputs $\{o_1, o_2, \ldots, o_G\}$ from the current policy $\pi_\theta$. The training objective is to maximize the following surrogate loss:

$$\mathcal{L}_{grpo} = \mathbb{E}\left[\frac{1}{G}\sum_{g=1}^{G}\Big(\min\left(\rho_g A_g, \text{clip}(\rho_g, 1-\epsilon, 1+\epsilon)A_g\right)\right. \quad (3)$$
$$\left.- \beta\mathbb{D}_{KL}(\pi_\theta\|\pi_{ref})\Big)\right],$$

where $\rho_g = \frac{\pi_\theta(o_g|q)}{\pi_{old}(o_g|q)}$ is the importance sampling ratio, and $\pi_{ref}$ is the reference model (the cold-start checkpoint in our work). The advantage $A_g$ is computed by normalizing the rewards within the group: $A_g = \frac{r_g - \text{mean}(\mathbf{r})}{\text{std}(\mathbf{r})}$.

While conventional reinforcement learning relies on format and accuracy-based rewards, the specialization of our model for 4D environments hinges upon our proposed Spatiotemporal Reward (ST-Reward). This reward mechanism serves as a critical supervisor, grounding the ST-CoT reasoning trajectories in precise spatial and temporal physical quantities to ensure logical consistency. As illustrated in Fig. 4, the integration of ST-Reward facilitates the emergence of sophisticated 4D analysis by refining the policy's ability to generate physically-grounded thoughts.

**Accuracy and Format Reward.** The accuracy reward $R_{Acc}$ evaluates the correctness of the final prediction, reinforcing alignment with gt labels: $R_{Acc} = 1$, if answer is right; otherwise, $R_{Acc} = 0$. The format reward complements the accuracy reward by enforcing strict adherence to the predefined response structure. We use regular expressions to verify that the generated trajectories follow the structure:

```
<thinking> Textual Reasoning...
    Object Center:[...]
    Camera Center:[...]
    Textual Reasoning...</thinking>
    <answer>Final Answer</answer>.
```

The total format reward is decomposed as:

$$R_{Fmt} = \lambda_1 R_{Stru\_fmt} + \lambda_2 R_{ST\_fmt}, \quad (4)$$

where $R_{Stru\_fmt} \in \{0, 1\}$ indicates whether the response is correctly encapsulated within <thinking> and

| Models | MLLM4D-Bench | | | | | | Avg. |
|---|---|---|---|---|---|---|---|
| | **Camera** | | **Object** | **Object & Camera** | | | |
| | Abs. Dis | Rel. Dir. | Abs. Dis. | Abs. Dis. | Rel. Dis. | Rel. Dir. | |
| Human | 88.2 | 86.5 | 87.8 | 86.9 | 87.1 | 87.3 | 87.3 |
| *Proprietary Models* | | | | | | | |
| GPT-4o (Hurst et al., 2024) | 34.8 | 57.6 | 32.7 | 36.7 | 56.7 | 51.0 | 44.9 |
| Gemini2.5-Flash (Comanici et al., 2025) | 35.2 | 59.4 | 24.1 | 37.8 | 55.3 | 50.6 | 43.8 |
| Gemini2.5-Pro (Comanici et al., 2025) | 37.4 | 57.5 | 31.1 | 38.6 | 60.1 | 55.0 | 46.6 |
| *Open-Source Models* | | | | | | | |
| Qwen2.5-VL-7B (Bai et al., 2025b) | 25.0 | 58.0 | 24.6 | 25.0 | 42.5 | 30.6 | 34.3 |
| Qwen2.5-VL-32B (Bai et al., 2025b) | 32.2 | 52.0 | 31.0 | 30.0 | 3.2 | 9.1 | 26.2 |
| Qwen3-VL-8B (Bai et al., 2025a) | 36.0 | 60.4 | 33.8 | 27.2 | 26.4 | 28.2 | 35.3 |
| Qwen3-VL-32B (Bai et al., 2025a) | 40.4 | 60.8 | 35.1 | 35.2 | 39.5 | 37.0 | 41.3 |
| LLaVA-NeXT-Video-7B (Zhang et al., 2024) | 22.6 | 26.0 | 24.6 | 24.3 | 33.4 | 32.6 | 27.3 |
| InternVideo2.5-8B (Wang et al., 2025e) | 9.4 | 48.2 | 9.5 | 9.3 | 22.3 | 18.6 | 19.6 |
| InternVL2.5-8B (Chen et al., 2024) | 27.5 | 43.4 | 5.8 | 7.2 | 21.0 | 16.1 | 20.2 |
| InternVL2.5-38B (Chen et al., 2024) | 7.7 | 16.4 | 9.4 | 14.1 | 24.0 | 17.4 | 14.8 |
| InternVL3.5-8B (Wang et al., 2025d) | 34.5 | 55.9 | 33.7 | 34.3 | 41.2 | 36.0 | 39.3 |
| InternVL3.5-38B (Wang et al., 2025d) | 24.3 | 22.6 | 24.0 | 40.9 | 40.7 | 37.1 | 31.6 |
| *3D Spatial Reasoning Models* | | | | | | | |
| VLM-3R (LLa.-Video-7B) (Fan et al., 2025) | 29.0 | 56.0 | 24.6 | 29.2 | 20.4 | 24.6 | 30.6 |
| VG-LLM (Qwen2.5-VL-7B) (Zheng et al., 2025) | 54.9 | 55.7 | 55.6 | 55.7 | 61.8 | 54.3 | 56.3 |
| **Our MLLM-4D (Qwen2.5-VL-7B)** | 73.3 | 68.1 | 75.7 | 73.0 | **70.9** | 60.4 | 70.2 |
| **Our MLLM-4D (Qwen3-VL-8B)** | **73.4** | **71.9** | **76.3** | **74.3** | 69.2 | **70.9** | **72.7** |

*Table 1.* Comparison of different models on MLLM4D-Bench. Bold font indicates the best performance.

<answer> tags, fostering a standardized CoT workflow. More importantly, we introduce $R_{ST\_fmt}$ rewards the explicit provision of Object Center and Camera Center coordinates in bracketed array formats. This structural constraint serves as a prerequisite, ensuring the reasoning process is parsable for subsequent spatiotemporal reward computation.

**Spatiotemporal Reward.** To constrain reasoning trajectories within a physically plausible 4D world state, we introduce the ST-reward. This reward ensures the model's internal reasoning is grounded in a physically plausible 4D world state rather than mere 2D pixel displacement. It quantitatively evaluates the model's ability to localize the camera and object at critical temporal anchors, specifically the start and end frames of the video sequence.

The camera and object rewards are computed by mapping the Mean Euclidean Error (MEE) between predicted coordinates $\mathbf{p}_i$ and ground-truth centers $\mathbf{g}_i$ to a normalized range $[0, 1]$ via an exponential decay function:

$$R_{Cam/Obj} = \exp(-\frac{1}{n}\sum_{i=1}^{n}||\mathbf{p}_i - \mathbf{g}_i||_2) \quad (5)$$

where $||\cdot||_2$ denotes the $L_2$ norm. A reward of 1.0 indicates perfect spatial-temporal alignment. Thus the final ST-reward is a composite signal defined as:

$$R_{ST} = \lambda_{Cam}R_{Cam} + \lambda_{Obj}R_{Obj}. \quad (6)$$

By enforcing explicit coordinate prediction, $R_{ST}$ serves as a physical regularizer, effectively mitigating spatiotemporal hallucinations and ensuring the reasoning process adheres to the underlying 4D dynamics. Consequently, the total reward $R$ for GRPO training is defined as:

$$R = \lambda_{Acc}R_{Acc} + \lambda_{Fmt}R_{Fmt} + \lambda_{ST}R_{ST}. \quad (7)$$

## 5. Experiments

*Please refer to the Appendix for the experimental setup, such as implementation details, and the comparison baselines.*

### 5.1. Comparisons on MLLM4D-Bench.

We compare our MLLM-4D with baselines for 4D spatial-temporal reasoning on MLLM4D-Bench. As shown in Table 1, MLLM-4D significantly outperforms all proprietary models, open-source and 3D spatial reasoning MLLMs. Specifically, our MLLM-4D (Qwen3-VL-8B) variant achieves a state-of-the-art average score of 72.7%, surpassing high-performing proprietary models like Gemini 2.5 Pro (46.6%) by a substantial margin. Qualitative comparisons provided in Fig. 12, Fig. 13 and Fig. 14 further highlight our model's superiority. Beyond simply providing accurate final answers, MLLM-4D demonstrates interpretable 4D reasoning by explicitly modeling the Spatiotemporal Chain-of-Thought (ST-CoT). In contrast, baseline models, including open-source models like Qwen3-VL-8B and 3D spatial reasoning models like VG-LLM, often lack a fun-

| Models | Real | Synthetic | Overall |
|---|---|---|---|
| Human | 99.7 | 96.2 | 98.8 |
| *Proprietary Models* | | | |
| GPT-4o | 60.0 | 49.9 | 57.5 |
| Gemini-2.5-Pro | 63.5 | 57.3 | 62.0 |
| *Open-Source Models* | | | |
| Qwen2.5-VL-7B | 43.3 | 45.6 | 43.8 |
| Qwen2.5-VL-72B | 53.1 | 52.6 | 53.0 |
| Qwen3-VL-8B | 52.1 | 52.4 | 52.2 |
| InternVideo2.5-8B | 52.7 | 44.5 | 50.7 |
| LLaVA-NeXT-Video-7B | 38.2 | 29.9 | 36.2 |
| *3D Spatial Reasoning Models* | | | |
| VLM-3R (LLaVA-NeXT-Video-7B) | $36.9_{0.9\downarrow}$ | $24.7_{5.2\downarrow}$ | $33.9_{2.3\downarrow}$ |
| VG-LLM (Qwen2.5-VL-7B) | $49.5_{6.2\uparrow}$ | $37.3_{8.3\downarrow}$ | $46.5_{2.7\uparrow}$ |
| **Our MLLM-4D (Qwen2.5-VL-7B)** | $59.4_{16.1\uparrow}$ | $49.7_{4.1\uparrow}$ | $57.0_{13.2\uparrow}$ |
| **Our MLLM-4D (Qwen3-VL-8B)** | $63.1_{11.0\uparrow}$ | $54.4_{2.0\uparrow}$ | $61.0_{8.8\uparrow}$ |

*Table 2.* Evaluation on VLM4D benchmark.

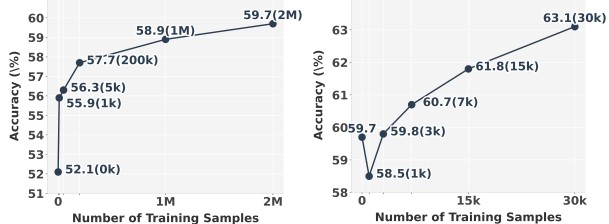

*(a)* SFT training data scaling.    *(b)* RFT training data scaling.
*Figure 5.* Scalability of training data on SFT and RFT stage.

damental understanding of 4D spatiotemporal dynamics. These baselines rely on guesswork or 2D visual cues, leading to incorrect results in 4D reasoning tasks.

### 5.2. Comparisons on VLM4D Benchmark.

To demonstrate the generalizability of our model on out-of-distribution datasets, we further evaluate performance on VLM4D benchmark (Zhou et al., 2025b). As shown in Table 2, MLLM-4D maintains superior performance, outperforming based model (Qwen3-VL-8B and Qwen2.5-VL-7B) and specialized 3D spatial reasoning models (VLM-3R and VG-LLM) by a significant margin. Qualitative results provided in Fig. 15 to Fig. 17 illustrate that by leveraging the ST-CoT reasoning paradigm, the 4D spatiotemporal intelligence of MLLM-4D can effectively translate to diverse environments beyond our training distribution.

### 5.3. Ablation Studies

**Scalability of Training Data.** To evaluate the data scalability of MLLM4D-2M, we evaluate performance across training subsets of 10K, 50K, 200K, 1M, and 2M QA pairs using real-world videos of the VLM4D benchmark. As illustrated in Fig. 5 (a), the model exhibits a rapid performance surge in the initial scaling phase, with accuracy rising from a 52.1% baseline to 57.7% when using 200K samples. Beyond this threshold, performance continues to scale consistently, reaching 59.7% at 2M samples. This sustained upward trajectory confirms that the MLLM4D-2M dataset effectively captures increasingly complex 4D spatial-temporal patterns as the data volume expands. We further analyze the scaling behavior during the RFT stage using subsets of MLLM4D-R1-30k ranging from 1K to 30K

| Models | MLLM4D-Bench | VLM4D |
|---|---|---|
| Baseline | 35.3 | 52.1 |
| SFT (w/o data in Sec. 3) | 59.9 | 56.2 |
| SFT (w/ data in Sec. 3)) | 70.1 | 59.7 |
| GRPO (w/o ST-reward) | 70.5 | 61.4 |
| GRPO (w/ ST-reward) | 72.7 | 63.1 |

*Table 3.* Ablation study of MLLM-4D framework.

pairs. As shown in Fig. 5 (b), despite a minor initial fluctuation at 1K due to the model adapting to the reasoning format, the model exhibits a robust scaling trend beyond 3K samples. Accuracy improves from 59.8% to a peak of 63.1% at 30K samples, highlighting the scalability of our MLLM4D-R1-30k dataset.

**Effectiveness of MLLM-4D.** Based on the ablation study presented in Table 3, our post-training framework progressively enhances 4D spatiotemporal reasoning capabilities of MLLM across both the MLLM4D-Bench and VLM4D benchmarks. The results highlight the following key insights: (1) *Impact of SFT*: the transition from the Baseline to SFT version of MLLM-4D yields a substantial performance leap, particularly on the MLLM4D-Bench, where scores nearly double from 35.3% to 70.1%. This confirms that the MLLM4D-2M dataset effectively establishes a robust foundational 4D understanding. (2) *Scalable Spatialtemporal Data Curation*: we compared our primary data curation pipeline against an alternative method based on monocular videos (see details in Appendix 7.5). While both large-scale 4D datasets drive significant improvements, the pipeline proposed in Sec. 3 yields superior results. This suggests that our pipeline provides higher-quality data, whereas monocular-based pipelines often suffer from depth ambiguity and diminished spatial accuracy. (3) *Benefits of RFT*: applying GRPO with standard accuracy and format rewards provides robust performance gains over SFT. (4) *Advantage of ST-reward*: the full post-training, including the ST-reward, achieves the highest performance (72.7% on MLLM4D-Bench and 63.1% on VLM4D). This validates that grounding the reasoning process in 4D physical quantities through ST-reward functions significantly boosts the model's capacity for complex spatiotemporal reasoning.

## 6. Conclusion

We present MLLM-4D, a comprehensive framework that advances the 4D spatiotemporal reasoning capabilities of MLLMs. To address the scarcity of high-quality data, we introduce an automated curation pipeline for large-scale 4D instructional pairs. We bridge the gap in specialized 4D-aware modeling by establishing foundational 4D understanding via SFT and subsequently unlocking advanced 4D reasoning capabilities by employing GRPO with specialized ST-CoT and ST-reward functions. We hope that our data, models, and methodology inspire future research on 4D spatiotemporal intelligence and facilitate the development of interactive AI systems in real-world environments.

## Acknowledgement

This work was financially supported in part by the Science and Technology Development Fund, Macau SAR, under Grant 0193/2023/RIA3 and 0079/2025/AFJ, the University of Macau under Grant MYRG-GRG2024-00065-FST-UMDF, the National Natural Science Foundation of China (Project No. 62506064) and Guangdong Provincial Regional Joint Fund (Project No. 2024A1515110052). The computational resources are supported by SongShan Lake HPC Center (SSL-HPC) in Great Bay University.

## Impact Statement

MLLM-4D provides a comprehensive framework designed to boost the spatial-temporal intelligence of MLLMs. This advancement is particularly beneficial for interactive AI systems such as VR/AR, autonomous driving, and robotics. In varied application scenarios, it is essential to follow the corresponding usage guidelines to ensure its proper and ethical application, minimizing any potential risks.

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

# 7. Appendix

## 7.1. Implementation Details

**Training Details.** MLLM-4D is built on Qwen3-VL-8B-Instruct (Bai et al., 2025a) and Qwen2.5-VL-7B-Instruct (Bai et al., 2025b). During training, we utilize LoRA (Hu et al., 2022) configured with an update matrix rank of 128 and an adaptation scaling parameter of 256, and we limit video frames to 32. In the SFT stage, we train the model using Adam optimizer for one epoch ($30k$ steps). We employ a cosine learning rate schedule with a peak learning rate of $2 \times 10^{-5}$, a warmup ratio of 0.01, and a global batch size of 8. In the cold start stage, we use a similar setting as in the SFT stage to train the model for about 200 steps. In the RFT stage, we perform 12 rollouts per question and set the default sampling temperature to 1. The $\lambda_{Acc}, \lambda_{Fmt}, \lambda_{ST}, \lambda_1, \lambda_2, \lambda_{Cam}, \lambda_{Obj}$ are set to 0.5, 0.2, 0.3, 0.5, 0.5, 0.5, 0.5, respectively. We train the model for $15k$ steps with a KL divergence coefficient $\beta$ of 0.1 and a learning rate of $5 \times 10^{-5}$. All experiments were conducted on 8 H100 80G GPUs; the training takes 12 hours for the SFT and cold start stage, and 50 hours for the RFT stage.

**Comparison Baselines.** We compare our MLLM-4D with a comprehensive suite of state-of-the-art MLLMs. For proprietary models, we include GPT-4o (Hurst et al., 2024), Gemini2.5-Flash (Comanici et al., 2025) and Gemini2.5-Pro (Comanici et al., 2025). Regarding open-source MLLMs, we consider various scales of Qwen2.5-VL (Bai et al., 2025b), Qwen3-VL (Bai et al., 2025a), LLaVA-NeXT-Video (Zhang et al., 2024), InternVideo2.5 (Wang et al., 2025e), InternVL2.5 (Chen et al., 2024), and InternVL3.5 (Wang et al., 2025d) series. We also include specialized 3D spatial reasoning models, such as VG-LLM (Zheng et al., 2025) and VLM-3R (Fan et al., 2025).

## 7.2. Details of MLLM4D-2M and MLLM4D-R1-30k Dataset Construction

**Moving Object Noun categories extraction.** Given the video sequence $\{I_i\}_{i=1}^K$, we employ a video MLLM, *i.e.* Gemini-2.5-flash (Comanici et al., 2025) using the prompt shown in Fig. 6, to identify all moving entities and extract their corresponding noun categories.

**QA Pair Generation.** We organize the extracted information into a structured meta-data format. This encompasses per-frame camera poses $\{C_i\}_{i=1}^K$, where $C_i = [R_i|t_i]$ consists of a $3 \times 3$ rotation matrix $R_i$ and a $3 \times 1$ translation vector $t_i$; object-level metric 3D points $\{P_{i,m}\}_{i=1,...,K;m=1,...,M}$; and their corresponding semantic descriptions $\{T_m\}_{m=1}^M$.

Using ***physical-based spatial-temporal computation***, we subsequently generate QA pairs of different tasks across several spatiotemporal reasoning tasks:

**User Prompt:**
You are a specialized video entity recognizer. Your role is to accurately identify and compile "active" entities from the video. An "active" entity is any visible object that shows dynamic behaviors, such as movement, action, or interaction with other objects. Ignore background elements and any inactive entities.

⚠️ Output Requirement:
1. Every entity must include a description.
3. Return the results strictly in format as follows:

```
{
    "active": [
        {
            "en": "entity name (e.g., man)",
            "cat": "broad category (e.g., person, vehicle, animal, or other appropriate category)",
        },
        {
            "en": "entity name (e.g., gril)",
            "cat": "broad category (e.g., person, vehicle, animal, or other appropriate category)",
        }
    ]
}
```

*Figure 6.* Illustration of the prompt used to identify all moving entities and extract their corresponding noun categories.

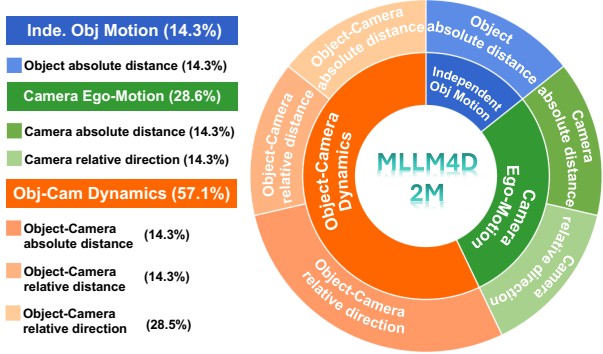

*Figure 7.* The components of our MLLM4D-2M.

• *Camera Absolute Distance*: Measures the movement of camera between any two frames $i$ and $j$. We first compute the camera center in world coordinates as $Center_i = -R^T t$. Then the absolute distance $d_{cam\_abs\_dis}$ is the $L_2$ norm: $d_{cam\_abs\_dis} = \|Center_j - Center_i\|_2 = \sqrt{\sum_{k=1}^3 (Center_{j,k} - Center_{i,k})^2}$. Question template: "Approximately how far (in meters) did the camera move between $<$ frame $i >$ and $<$ frame $j >$?"

• *Camera Relative Direction*: Determines movement relative to the camera's orientation at frame $i$ between any two frames $i$ and $j$. we first compute the the world-space displacement $\Delta t = t_j - t_i$. To determine the movement relative to the camera's perspective at frame $i$, we project this vector into the local camera coordinate system using the transpose of the rotation matrix $R_i^T$: $d_{cam\_rel\_dir} = R_i^T \cdot \Delta t$. The resulting vector $d_{cam\_rel\_dir} = [dx, dy, dz]$ is interpreted via standard camera conventions, where $+X, +Y, +Z$ correspond to right, down, and forward direction, respectively. Question template: "During the sequence between $<$ frame $i >$ and $<$ frame $j >$, what was the primary consistent translation of the camera's

movement relative to its position at the start?"

- *Object Absolute Distance*: Computes the movement distance of an object between any two frames $i$ and $j$. We define the object centroid at frame $i$ as $\bar{P}_i = \frac{1}{M}\sum_{m=1}^{M} P_{i,m}$. Then the object absolute distance $d_{obj\_abs\_dis}$ is calculated as the $L_2$ norm of the vector connecting their centroids: $d_{obj\_abs\_dis} = \|\bar{P}_j - \bar{P}_i\|_2 = \sqrt{\sum_{k=1}^{3}(\bar{P}_{j,k} - \bar{P}_{i,k})^2}$. Question template: "Approximately how far (in meters) did $<$ *object m description* $>$ move between $<$ `frame` $i$ $>$ and $<$ `frame` $j$ $>$?"

- *Object-Camera Absolute Distance*: Computes the proximity of an object at frame $j$ to the camera at frame $i$. We first compute the camera center in world coordinates as $Center_i = -R^T t$. Then the absolute distance $d_{obj}$ between the camera and the object is defined as the minimum distance from the camera center to any point within the object's point set: $d_{obj\text{-}cam\_abs} = \min \|P_{j,m} - Center_i\|_2$. Question template: "What is the approximate distance (in meters) between the camera (or the observer filming) in $<$ `frame` $i$ $>$ and the nearest point of the $<$ `object m description` $>$ in $<$ *frame j* $>$?"

- *Object-Camera Relative Distance*: Determines whether the camera and object are converging or diverging between any two frames $i$ and $j$. We first find the camera center to any point within the object's point set: $d_i = \min \|P_{i,m} - Center_i\|_2$. Then the relative change in distance, $\Delta d$, is the difference between the instantaneous distances at frame $j$ and frame $i$: $\Delta d = d_j - d_i$. Based on a distance threshold $\tau$, the relative movement is classified into a discrete state $S = $ farther if $\Delta d > \tau$; closer if $\Delta d < -\tau$; otherwise not moving. Question template: "During the sequence between $<$ `frame` $i$ $>$ and $<$ `frame` $j$ $>$, is the distance between $<$ `object m description` $>$ and the camera (or the observer filming) getting closer or farther away?"

- *Object-Camera Relative Direction*: determine the relative direction between the camera and the moving object between is getting left/right or closer/farther between any two frames $i$ and $j$. We first transform the object points from both time steps into the camera's local coordinate system at frame $i$: the local camera coordinates: $P_i^c = R_i P_i + t_i$. Then we calculate the centroid of the object in the camera's coordinate space for both sets: $\bar{P}_i^c = \frac{1}{M}\sum_{n=1}^{M}(R_i P_{i,m} + t_i)$, $\bar{P}_j^c = \frac{1}{M}\sum_{m=1}^{M}(R_i P_{j,m} + t_i)$. We then classify the lateral change $\Delta x = \bar{P}_{j,x}^c - \bar{P}_{i,x}^c$ (left/right) and the longitudinal change $\Delta z = \bar{P}_{j,z}^c - \bar{P}_{i,z}^c$ (closer/farther) us-

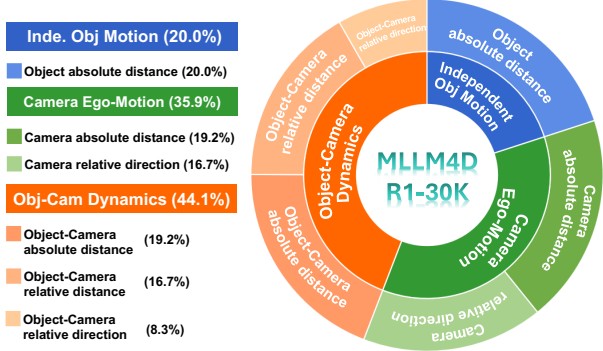

*Figure 8.* The components of our MLLM4D-R1-30k.

ing threshold $\tau$. Question template: "During the sequence between $<$ `frame` $i$ $>$ and $<$ `frame` $j$ $>$, is $<$ `object m description` $>$ getting left or right from the camera (or the observer filming) relative to camera's position at the start?" or "During the sequence between $<$ `frame` $i$ $>$ and $<$ `frame` $j$ $>$, is $<$ `object m description` $>$ getting closer or farther away from the camera (or the observer filming) relative to camera's position at the start?"

**Data Filtering and Balancing Protocols.** After calculating the ground-truth values, we apply templates to formulate the final QA pairs. We implement several filtering and balancing protocols to ensure quality: we limit the number of QA pairs per video to maintain scene diversity and shuffle multiple-choice options to eliminate positional bias. Additionally, the distractors for numerical options are randomly generated within 25%–175% of the true value to prevent unrealistic shifts. After filtering, we retain $2M$ high-quality QA pairs across approximately $100k$ videos, forming the large-scale *MLLM4D-2M* dataset for supervised fine-tuning.

### 7.3. Details of Cold Start

To align the model's output with the desired reasoning format, we conducted a brief cold-start fine-tuning phase consisting of 200 steps before GRPO training, following the same hyperparameters as SFT. The cornerstone of this phase involves constructing a reasoning dataset with CoT annotations derived from pre-collected question-answer pairs. The construction process is detailed as follows:

**Subset Sampling.** We begin by sampling a subset $\mathcal{D}_0 = \bigcup_{s=1}^{7} \mathcal{D}_s$ from the MLLM4D-2M dataset, which is constructed by drawing samples across all 7 distinct scenarios covered in the original dataset. Specifically, $\mathcal{D}_s = \{\mathcal{I}_s^i\}_{i=1}^{N} = \{\langle \mathcal{Q}_s^i, \mathcal{A}_s^i, \mathcal{V}_s^i \rangle\}_{i=1}^{N}$, where each instance is uniformly sourced from the diverse scenario pool.

**Multi-path CoT Generation.** For each sample $\mathcal{I}_s^i \in \mathcal{D}_0$, we utilize Gemini2.5-Pro (Comanici et al., 2025) to gen-

**Algorithm 1** Scenario-Adaptive Cold-Start Construction

---

1: **Input:** Initial subset $\mathcal{D}_0 = \bigcup_{s=1}^{7} \mathcal{D}_s$, Model $\mathcal{M}$, Reward function $R(\cdot)$
2: **Output:** Filtered cold-start dataset $\mathcal{D}_{cold}$
3: $\mathcal{D}_{cold} \leftarrow \emptyset$
4: **for** each scenario $s \in \{1, \ldots, 7\}$ **do**
5:     {Step 1: Multi-path Generation}
6:     **for** each sample $\mathcal{I}_s^i \in \mathcal{D}_s$ **do**
7:         Get K paths: $\{\langle \hat{\mathcal{T}}_s^{i,k}, \hat{\mathcal{A}}_s^{i,k}, r_s^{i,k} \rangle\}_{k=1}^{K}$ using $\mathcal{M}$
8:     **end for**
9:     {Step 2: Scenario-specific Filtering}
10:     Compute scenario mean reward as threshold:
        $\tau_s \leftarrow \frac{1}{N}\sum_{i=1}^{N} \hat{r}_s^i = \frac{1}{N}\sum_{i=1}^{N} \arg\max_k r_s^{i,k}$
11:     **for** each sample $i$ and path $k$ in scenario $s$ **do**
12:         **if** $r_s^{i,k} \geq \tau_s$ and $r_s^{i,k} > 0$ **then**
13:             $\mathcal{D}_{cold} \leftarrow \mathcal{D}_{cold} \cup \{\langle \mathcal{I}_s^i, \hat{\mathcal{T}}_s^{i,k}, \hat{\mathcal{A}}_s^{i,k} \rangle\}$
14:         **end if**
15:     **end for**
16: **end for**
17: **Return:** $\mathcal{D}_{cold}$

---

erate K independent reasoning processes $\hat{\mathcal{T}}_s^{i,k}$ and corresponding answers $\hat{\mathcal{A}}_s^{i,k}$. We then compute the reward $r_s^{i,k} = \text{Reward}(\hat{\mathcal{A}}_s^{i,k}, \mathcal{A}_s^i)$ for each reasoning-answer pair, where $\text{Reward}(.,.)$ is the reward function described in Sec 4.3. Consequently, we obtain a set of outputs $\mathcal{O}_s^i = \{\langle \hat{\mathcal{T}}_s^{i,k}, \hat{\mathcal{A}}_s^{i,k}, r_s^{i,k} \rangle\}_{k=1}^{K}$ for each $\mathcal{I}_s^i \in \mathcal{D}_0$.

**Scenario-specific Filtering.** Since the generative quality of Gemini2.5-Pro (Comanici et al., 2025) may vary across different data distributions, applying a global reward threshold can lead to an imbalance across scenarios. To mitigate this, we adopt a scenario-specific filtering strategy. For each sample $\mathcal{I}_s^i \in \mathcal{D}_0$, we first identify the output with the highest reward, denoted as $\hat{r}_s^i = \arg\max_k r_s^{i,k}$. We then compute a scenario-dependent threshold $\tau_s$ by averaging the maximum rewards within each scenario:

$$\tau_s = \frac{1}{N}\sum_{i=1}^{N} \hat{r}_s^i, \quad s \in 1, \ldots, 7. \qquad (8)$$

An item $\mathcal{I}_s^i$ is preserved in the final cold-start set if and only if $\hat{r}_s^i \geq \tau_s$. This strategy ensures that the model learns from the top-performing generations relative to each scenario's complexity, maintaining a balanced representation of all 7 scenarios in the reasoning dataset. This rule preserves approximately the top 50% of generations per question type while discarding degenerate (zero-reward) outputs. In practice, we set N = 2000 and K = 3, and finally, we get 7000 samples in the cold start set. We provide a pseudocode for this process in Algorithm 1.

**Other Details.** Fig. 10 presents the prompts used in the SFT, Cold Start, and GRPO stages. For the SFT stage, we

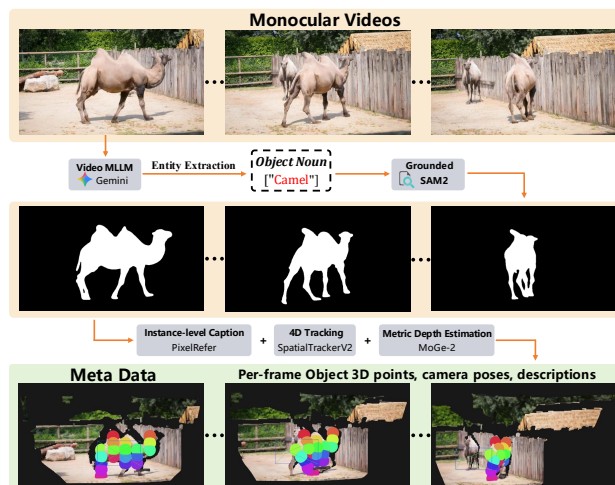

*Figure 9.* An alternative data pipeline based on monocular videos.

adopt the default system prompt of Qwen3-VL (Bai et al., 2025a), namely, "You are a helpful assistant." In the Cold Start and GRPO stage, we use specially designed system prompts to guide the model's output of 4D information.

### 7.4. Extended Evaluation on Several Heterogeneous Benchmarks

We conduct further comparisons across several heterogeneous benchmarks: spatial intelligence benchmarks such as MMSI-Bench (Yang et al., 2026a), OSI-Bench (Wu et al., 2025b), and ERQA (Team et al., 2025); and general-purpose benchmark such as Video-MME (Fu et al., 2025). We report the overall performance comparison and the 4D-related subsets accuracy of each benchmark, such as Motion camera and Motion object subset of MMSI-Bench, Dynamic Metric subset of OSI-Bench, Trajectory Reasoning and Action Reasoning subset of ERQA. As shown in Tab. 4, the consistent improvements across heterogeneous tasks and external benchmarks confirm that our MLLM-4D learns fundamental, transferable representations rather than merely overfitting to pipeline-specific heuristics.

### 7.5. Details of data curation pipeline based on monocular videos.

We also implement an alternative pipeline based on monocular videos, as shown in Fig. 9. First, we employ Gemini-2.5-flash, to identify all moving entities and extract their corresponding noun categories. We then utilize GroundedSAM2 for instance segmentation and tracking, yielding temporally consistent 2D masks across the sequence. These semantic descriptions are further enriched using PixelRefer. We sample pixels in each region and apply a 4D tracking method, such as SpatialTrakerV2 (Xiao et al., 2025) to track points in 4D space. Since 4D tracking method typically

| Models | MMSI-Bench | OSIBench | ERQA | Video-MME |
|---|---|---|---|---|
| Qwen2.5-VL | 27.2 / 27.6 | 15.4 / 22.9 | 40.6 / 42.3 | 55.4 |
| Qwen3-VL | 27.3 / **30.7** | 27.7 / 33.6 | 38.4 / 42.0 | 58.5 |
| VLM-3R | 24.0 / 27.9 | 22.8 / 25.3 | 30.4 / 31.8 | 53.3 |
| Spatial-MLLM | 16.0 / 25.8 | 13.2 / 13.3 | 36.2 / 35.8 | 44.9 |
| VG-LLM | 20.1 / 27.5 | 11.1 / 20.3 | 39.3 / 39.3 | 59.5 |
| Our MLLM-4D | **31.0** / 29.6 | **30.2** / **35.6** | **42.1** / **42.5** | **62.4** |

*Table 4.* Comparison on heterogeneous Benchmarks (Accuracy of 4D-specific subsets / Overall Accuracy).

| Models | 32 | 16 | 8 |
|---|---|---|---|
| Qwen3-VL | 52.2 | 54.3 | 53.3 |
| VLM-3R | 33.9 | 34.5 | 34.5 |
| VG-LLM | 46.5 | 47.1 | 46.7 |
| Our MLLM-4D | 61.0 | 61.3 | 61.0 |

*Table 5.* Sensitivity to the frame sampling.

produces depth at a relative-scale, we incorporate a metric-scale depth estimation method, such as MoGe-2 (Wang et al., 2025c), to align the final per-frame object-level points. This monocular-based pipeline often faces challenges with depth ambiguity and diminished spatial accuracy inherent in 4D tracking and monocular depth estimation.

### 7.6. Further Analysis of Sensitivity to Frame Sampling

We conduct frame sensitive experiment by reducing the input frames to each model to 16 and 8 on VLM4D. As shown in Tab. 5, our MLLM-4D and baseline methods are robust to frame sampling.

### 7.7. Limitations and Future Work

Although MLLM-4D demonstrates strong viusal-based spatial-temporal intelligence compared to existing MLLMs across various 4D reasoning tasks, it still faces constraints in processing long-duration video sequences. Due to the inherent input length limitations of current architectures, our model relies on frame sampling. An compelling direction for the future work would lies in exploring long-context 4D spatiotemporal reasoning.

## Question

**Example:**

**Question:** "What is the approximate distance (in meters) between the camera (or the observer filming) in frame 14 and the nearest point of the A dolphin with a streamlined body, a prominent dorsal fin, and a tail fluke, swimming in water in frame 15 of 31?\nOptions:\nA. 2.5\nB. 12.7\nC. 15.2\nD. 9.6"

**Video Frames:**

Frame 1  Frame 2  Frame 3  Frame 4     Frame 29  Frame 30  Frame 31  Frame 32

## SFT Stage

**System Prompt:**
"You are a helpful assistant."

**User Prompt:**
"{**Video Frames**}+**<video>**These are frames of a video.\n"+{**Question**}+"\nAnswer with the option's letter **from the given choices directly**."

## Cold Start & GRPO Stage

**System Prompt:**
"You are a video analysis assistant. Your goal is to solve the user's question by performing a detailed spatial-temporal analysis of the video content.
The response must follow a strict structure:
1. **Reasoning Process**: Enclosed within **<thinking></thinking>** tags.
2. **Final Answer**: Enclosed within **<answer></answer>** tags.
3. **Internal Reasoning Requirements:**
Inside the <thinking> tags, you must explicitly document your perception of the physical state at the start and the end of the relevant video segment. Use the following structured format for Spatial State:
**Spatial State (Initial Frame):**
  - Camera Center: [x, y, z] or null
  - Object Center: [x, y, z] or null
**Spatial State (Final Frame):**
  - Camera Center: [x, y, z] or null
  - Object Center: [x, y, z] or null
If any specific value is unavailable or cannot be inferred, output `null`. Ensure the reasoning leads logically from these physical states to the final answer."

**User Prompt:**
"{**Video Frames**}+**<video>**\n"+{**Question**}+"\nOutput the thinking process in **<thinking></thinking>** and \n final answer in **<answer></answer>** tags."

*Figure 10.* Detailed system prompt and user prompt setting for our SFT, Cold Start and GRPO stage.

## Spatiotemporal Chain-of-Thought Generation Prompt

```
"""
You are a high-performance Spatio-Temporal AI. You are currently processing a video stream and your internal 3D perception engine is active.

## TASK
Analyze the visual movement in the video to answer the Question.
Your output must be a Chain of Thought (CoT) that prioritizes **visual narrative**. You must derive the answer solely from **observable evidence** in the video frames.

## INPUT DATA (FOR INTERNAL SENSOR REFERENCE)
1. **Question**: [QUESTION_PLACEHOLDER]
2. **Options**: [OPTIONS_PLACEHOLDER]
3. **Internal Target**: [GROUND_TRUTH_PLACEHOLDER]
4. **Sensor Log**: [PHYSICS_DATA_PLACEHOLDER]
   *(Note: This log contains raw Camera Center and Object Center. If solving this problem does not require some data, they are set 'null')*

## MANDATORY RULES
1. **Strict "Spatial State" Formatting**:
   You must provide a snapshot of your perception at the beginning and the end. If a value is missing, output `null`.
   **Spatial State (Frame X):**
   - Camera Center: [x, y, z] or null
   - Object Center: [x, y, z] or null

2. **Visual-Driven Reasoning (No Math in Text)**:
   * **PROHIBITED**: "Because the X-coordinate changed...", "Based on the provided sensor log...".
   * **REQUIRED**: Use optical flow cues. (e.g., "The target object expands in the field of view, indicating a decrease in relative distance," or "The parallactic shift of the background suggests the camera is translating right.")

3. **Evidence-Based Conclusion (NO GUESSING)**:
   * When stating the final answer, you must justify it by summarizing the **visual trajectory**.
   * **DO NOT** simply state the answer. You must say: "Given that we observed [specific visual event] during the sequence, the only plausible value among the options is [Answer]."

## COT STRUCTURE REQUIREMENT
- **Step 1: Objective**: Define the goal.
- **Step 2: Start Frame Perception**: Describe the initial visual scene and initial position. Follow with **Spatial State**.
- **Step 3: Temporal progression & Evidence Collection**: This is the most important part. Describe the sequence of movement.
  - *Example*: "From Frame 16 to 20, the entity's feet are seen moving in a walking gait, and its silhouette becomes larger against the static doorframe, confirming positive forward velocity."
- **Step 4: End Frame Perception**: Describe the final visual state. Follow with **Spatial State**.
- **Step 5: Final Synthesis & Verification**: Summarize the **total visual change** and explain why it leads to the chosen option. Select the option that best fits the visual evidence.

## OUTPUT FORMAT (JSON ONLY)
```json
{
  "chain_of_thought": "The objective is to... \\n\\nIn Frame [Start], visually, we see... [Initial Evidence].\\n**Spatial State (Frame [Start]):**\\n- Object Center: [ ... ]\\n- Camera Center: [ ... ]\\n\\nAnalyzing the temporal flow, the visual evidence shows [Specific Evidence 1 followed by [Specific Evidence 2]. We see the distance gap closing because [Visual Clue]...\\n\\nBy Frame [End], the visual evidence confirms [Final State]...\\n**Spatial State (Frame [End]):**\\n- Object Center: [ ... ]\\n- Camera Center: [ ... ]\\n\\nSynthesizing the motion trajectory: the [Object] moved [Direction/Relative Distance] while the Camera [Ego-motion]. This visual progression directly supports the choice of [Answer] as the most accurate estimation."
}
"""
```

*Figure 11.* Detailed structure of the Spatiotemporal Chain of Thought (ST-CoT) Generation Prompt. The prompt is color-coded into four functional modules: ■ **System Role & Task** defines the AI persona and core mission; ■ **Input Data & Mandatory Rules** enforces visual-driven reasoning and constraints; ■ **CoT Structure** provides the step-by-step requirements for the reasoning chain; ■ **Output Format** specifies the data placeholders and JSON schema.

**Video**

**Question:**                                                                                          **Right Answer: A**

" Approximately how far (in meters) did the camera move between frame 0 and frame 26 of 31? Options: A. 1.2 B. 0.4 C. 0.8 D. 1.6?  Options: A. 2.5 meters B. 12.7 meters C. 15.2 meters D. 9.6 meters"

**MLLM-4D:** "**<thinking>**The objective is to determine the approximate distance (in meters) the camera moved between frame 0 and frame 26. In Frame 0, visually, we see a scene dominated by a dark metal fence on the right, partially obscured by dense green foliage. A concrete pathway leads into the distance, flanked by more vegetation and a white wall on the left. A distant, indistinct vehicle is barely visible behind the fence. **Spatial State (Frame 0):** - Object Center: null - Camera Center: [-0.0133, 0.0034, 0.0322]** Analyzing the temporal flow, the visual evidence shows a continuous forward translation of the camera. **From frame 0 to 26, the foreground elements, such as the concrete steps and the base of the metal fence, progressively expand in the field of view, indicating a decrease in their relative distance. The fence itself appears to 'flow' horizontally across the screen from right to left, while simultaneously moving further into the background.** The distant white wall and the overall horizon line also exhibit a subtle but consistent recession, suggesting a sustained forward motion. The vegetation, initially closer, becomes more prominent and detailed as the frames progress, confirming the camera is advancing through the environment. By Frame 26, the visual evidence confirms that the camera has significantly advanced along the path. The concrete steps and the lower portions of the fence are now much larger and closer to the camera's viewpoint. The distant white wall has shifted further into the background, and the overall perspective of the scene has changed considerably, reflecting a notable forward displacement. **Spatial State (Frame 26):** - Object Center: null - Camera Center: [-0.0367, 0.0090, 1.1737]** Synthesizing the motion trajectory: the camera underwent a continuous forward translation, evidenced by the consistent expansion of foreground objects and the apparent recession of background elements. This visual progression directly supports the choice of 1.2 meters as the most accurate estimation for the camera's movement, as it represents a moderate yet discernible forward shift in the scene.**</thinking><answer>A</answer>**" ✅

**VG-LLM:** "From the given perspective, the object fence is to the right . It seems to the front. The observer may be about 1.6 meters from fence. The object tree is to the left . It seems to the front. The observer may be about 1.6 meters from tree. Based on these measurements, **the observer is approximately 0.8 meters away from the center of fence (red point) and 1.6 meters from the center of tree (blue point)**. The observed spacing is 0.8 meters.**<answer>0.8</answer>**" ❌

**Qwen3-VL:** "To estimate how far the camera moved between frame 0 and frame 26 of 31, we need to analyze the change in perspective or position of objects in the scene. Looking at the video frames from 0 to 26 (out of 31), the camera appears to be moving forward along a path, as indicated by the changing perspective of the fence, vegetation, and the ground. The camera is likely handheld and moving slowly, possibly walking or moving along a narrow path. Lets break it down: - The movement is not rapid, and the scene is relatively static, so the motion is likely due to the camera operator moving forward. - The camera's forward motion is gradual, and the distance covered is small — less than 2 meters, likely around 1 meter or less, given the slow pace and the nature of the environment (a narrow path with vegetation). Looking at the options: - A. 1.2 meters — plausible, but a bit high for such a slow movement. - B. 0.4 meters — too small; the camera clearly moves forward and changes perspective. - C. 0.8 meters — reasonable for a slow forward motion over 26 frames. - **D. 1.6 meters — too large for such a short duration (26/31 ≈ 84% of the video). Given the slow, steady movement and the gradual change in perspective, 0.8 meters is the most reasonable estimate. <answer> C. 0.8 </answer>**" ❌

*Figure 12.* Qualitative comparison on MLLM4D-Bench.

**Video**

**Question:**                                                                                          **Right Answer: D**

" Approximately how far (in meters) did the The woman is wearing a red jacket and black pants move between frame 14 and frame 29 of 31? Options: A. 1.7 meters  B. 1.0 meters C. 0.4 meters D. 1.3 meters"

**MLLM-4D:** "**\<thinking\>**The objective is to determine the approximate distance (in meters) the woman in the red jacket and black pants moved between frame 14 and frame 29.In Frame 14, visually, we see the woman in the red jacket and black pants positioned on the right side of the frame, walking towards the left. Her figure occupies a moderate portion of the visual field, indicating she is neither extremely close nor distant from the camera. She is moving past other attendees and towards the left edge of the scene. **\*\*Spatial State (Frame 14):\*\* - Object Center: [-0.4738, 0.5118, 2.3389] - Camera Center: [-0.0228, 0.0021, 0.0011] Analyzing the temporal flow, the visual evidence shows the woman in the red jacket and black pants continuously moving from right to left across the frame. As the sequence progresses, her apparent size in the frame gradually decreases, and she moves further into the scene, becoming less prominent. Her position shifts from the right-center towards the left-center, and eventually, she moves out of the frame entirely by frame 29. Concurrently, the background elements, such as the booth on the right and the distant wall, exhibit a subtle parallactic shift, suggesting a minor forward movement of the camera itself, but the primary motion is driven by the woman's traversal of the space.**By Frame 29, the visual evidence confirms that the woman in the red jacket and black pants has moved significantly further to the left, having largely exited the frame. Her figure is now much smaller and positioned towards the far left, indicating she has traversed a considerable distance within the environment. **\*\*Spatial State (Frame 29):\*\* - Object Center: [-1.1349, 0.5265, 2.3828] - Camera Center: [-0.0233, 0.0022, 0.0012]** Synthesizing the motion trajectory: the woman in the red jacket and black pants moved substantially from right to left across the visual field, decreasing in apparent size and moving deeper into the scene. This visual progression, where she traverses a significant portion of the frame and moves out of view, directly supports the choice of 1.3 meters as the most accurate estimation for the distance covered.**\</thinking\>\<answer\>D\</answer\>**" ✅

**VG-LLM:** "From the given data, The woman is 1.7 meters away from the start point (frame 14) and 1.7 meters from the end point (frame 29). **The distance between these two points is 0.0 meters. Therefore, the distance between the start and end points is 1.7 meters.** The correct answer is A. 1.7.**\<answer\>A\</answer\>**" ❌

**Qwen3-VL:** "To estimate the distance the woman in the red jacket and black pants moved between frame 14 and frame 29 of 31, we need to analyze the spatial progression of her position in the video frames. ### Step 1: Understand the context The video shows a crowded convention hall with people walking. The woman in the red jacket and black pants is walking from the right side of the frame toward the left, passing through the aisle. Her movement is consistent and directional. ### Step 2: Frame rate and time interval The video is 31 frames long, and we are analyzing the movement between frame 14 and frame 29. That's 15 frames (29 - 14 + 1 = 16 frames, but typically we consider the interval as 15 frames of motion between frame 14 and frame 29). Assuming a standard frame rate of 30 fps, this corresponds to approximately 0.5 seconds (15/30 = 0.5 seconds). ### Step 3: Estimate distance using visual cues We can use the known scale of the environment to estimate distance. For example, the booths and people in the background provide a reference. The woman walks past a booth on the right, then continues walking left. **Between frame 14 and frame 29, she moves from roughly mid-right to near the center of the frame, passing several people and booths. In a typical convention hall, a person walking at a normal pace covers about 1.5 to 2 meters per second. Over 0.5 seconds, that's approximately 0.75 to 1.0 meters.**### Final Answer:**\<answer\> B. 1.0 \</answer\>**" ❌

*Figure 13.* Qualitative comparison on MLLM4D-Bench.

**Video**

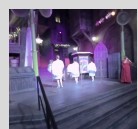 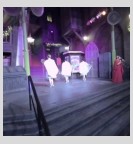 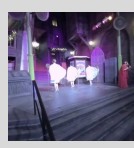 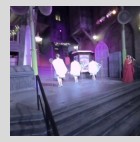 ... 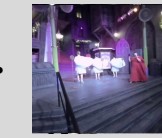 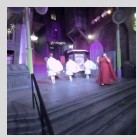 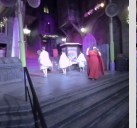 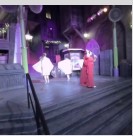

**Question:**                                                                                    **Right Answer: B**

" What is the approximate distance (in meters) between the camera (or the observer filming) in frame 4 and the nearest point of the The girl is wearing a white dress in frame 21 of 31? Options: A. 4.1 B. 5.4 C. 8.9 D. 6.8"

**MLLM-4D:** "**<thinking>**The objective is to determine the approximate distance (in meters) between the camera's position in Frame 4 and the nearest point of the girl wearing a white dress in Frame 21.In Frame 4, visually, we observe a stage performance. The camera is positioned at a low angle, looking up at the stage, which is elevated by a set of visible steps in the foreground. The girl wearing a white dress is situated in the mid-ground, towards the left side of the stage, standing behind a small, round object. Her figure appears relatively small within the overall stage composition, indicating she is at a moderate distance from the camera's viewpoint.**Spatial State (Frame 4):** - Object Center: [-0.8333, 0.3582, 5.6958]- Camera Center: [-0.0267, 0.0003, 0.0003] Analyzing the temporal flow from Frame 4 to Frame 21, the visual evidence shows that the camera maintains a remarkably stable position. There is no discernible ego-motion; the foreground steps and the overall stage structure remain fixed in the frame, indicating the camera is stationary. Concurrently, the girl in the white dress performs a dance, turning and moving her arms. Her apparent size in the frame remains consistent throughout the sequence, and her position relative to the stage elements and the background does not exhibit any significant change in depth or lateral shift. This visual progression confirms that the girl's distance from the camera's initial viewpoint has not altered substantially.** By Frame 21, the visual evidence confirms that the girl in the white dress is still in a similar position on the stage, maintaining her relative size and depth perception as observed in Frame 4. The camera's perspective is identical to that in Frame 4, with no observable change in the environment or the subject's apparent scale. **Spatial State (Frame 21):** - Object Center: [-0.8526, 0.3595, 5.7123 - Camera Center: [-0.0267, 0.0003, 0.0003]** Synthesizing the motion trajectory: the camera remained stationary, and the girl in the white dress maintained a consistent apparent size and relative position on the stage. This visual progression directly supports the choice of 5.4 as the most accurate estimation for the distance between the camera's position in Frame 4 and the girl's position in Frame 21. Given that the girl's apparent size and the camera's stable viewpoint suggest a moderate distance, and considering the options, 5.4 is the most plausible value.**</thinking>**<answer>B</answer>" ✅

**VG-LLM:** "From the given perspective, the distance of The girl is wearing a white dress is estimated at 4.1 meters. **<answer>A</answer>**" ❌

**Qwen3-VL:**" To estimate the distance between the camera (observer) and the nearest point of the girl wearing a white dress in frame 21 (which is the same as frame 4 since the video frames are sequential and the scene doesn't change), we need to analyze the spatial layout. Looking at frame 4, we can observe: - The stage is elevated with visible steps in the foreground.- The camera is positioned at the bottom of the stairs, looking up at the stage. The steps are visible in the foreground, and the girl is standing on the stage. From the perspective of the camera, the closest point on the stage to the camera is the front edge of the stage, which is at the bottom of the steps. The girl is standing further back on the stage. In frame 4, the girl is approximately 3-4 steps away from the front edge of the stage (assuming each step is about 0.25 meters high, the total height difference is around 1 meter). **However, since the camera is looking up at the stage, the actual distance is not just vertical but also includes horizontal distance. The stage appears to be about 2-3 meters wide from the front edge to where the girl is standing. The camera is positioned at the bottom of the stairs, and the stage is elevated. The horizontal distance from the camera to the front edge of the stage is approximately 2 meters**. The nearest point of the girl wearing a white dress in frame 4 is 6.8 meters. **<answer> D. 6.8 </answer>** ❌

*Figure 14.* Qualitative comparison on MLLM4D-Bench.

**Video**

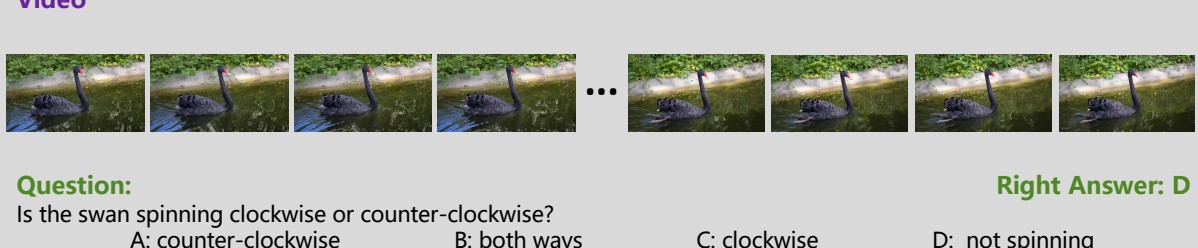

**Question:**                                                                                                **Right Answer: D**

Is the swan spinning clockwise or counter-clockwise?

A: counter-clockwise          B: both ways          C: clockwise          D:  not spinning

**MLLM-4D:** "**\<thinking\>**The objective is to determine if the swan is spinning clockwise or counter-clockwise based on visual evidence. In the initial frame, we observe a black swan with a prominent red beak, floating on the water. Its body is oriented towards the right side of the frame, with its head and neck extended. The background consists of a concrete bank and lush green foliage. The swan appears to be in a state of rest, with its wings folded against its body. **\*\*Spatial State (Initial Frame):\*\* - Object Center: [-0.0707, 0.0616, 1.6666] - Camera Center: [-0.0238, -0.0018, -0.0020] Analyzing the temporal flow, the visual evidence shows the swan maintaining a remarkably stable and consistent orientation throughout the sequence. Its head, neck, and body remain fixed in the same relative position within the frame, with no discernible rotation around its vertical axis. The swan's overall form and the pattern of its black feathers do not exhibit any visual cues of spinning. While the water ripples and the background elements like the concrete bank and foliage subtly shift due to the camera's own movement, the swan itself does not show any change in its rotational state. There is no visual indication of the swan's head turning to the left or right, or its body rotating in either direction.** By the final frame, the visual evidence confirms that the swan's orientation is virtually identical to the initial frame. Its head is still pointed towards the right, and its body maintains the same profile. The absence of any rotational movement of the swan's own form is clearly visible. **\*\*Spatial State (Final Frame):\*\* - Object Center: [-0.0589, 0.0532, 1.7008] - Camera Center: [-0.0237, -0.0017, -0.0019]** Synthesizing the motion trajectory: the swan's visual appearance and its orientation within the frame remained static throughout the sequence. There was no observable rotation of its body or head that would indicate a clockwise or counter-clockwise spin. This visual progression directly supports the choice of "not spinning" as the most accurate estimation.**\</thinking\>\<answer\>D\</answer\>**" ✅

**VG-LLM:** "The video shows the swan swimming in a circular motion, but it does not specify whether it is clockwise or counter-clockwise. **Therefore, the final answer is: B**" ❌

**Qwen3-VL:**" Observe the swan's movement in the video. The swan is moving slowly in the water, and its body is turning slightly to the right as it moves forward. This indicates a clockwise rotation. Step 2: Check for any counter-clockwise movement. There is no evidence in the video that the swan is turning counter-clockwise. Its motion is consistent with a gentle clockwise turn. Step 3: **Determine if the swan is spinning. The swan is not performing a rapid or full spin; it is moving forward while turning slightly. This is not spinning but rather a slow turn.** Step 4: Match the observation to the given options. The swan is turning clockwise, not counter-clockwise, not both ways, and it is not spinning. **Therefore, the final answer is: C**" ❌

*Figure 15.* Qualitative comparison on VLM4D benchmark.

**Video**

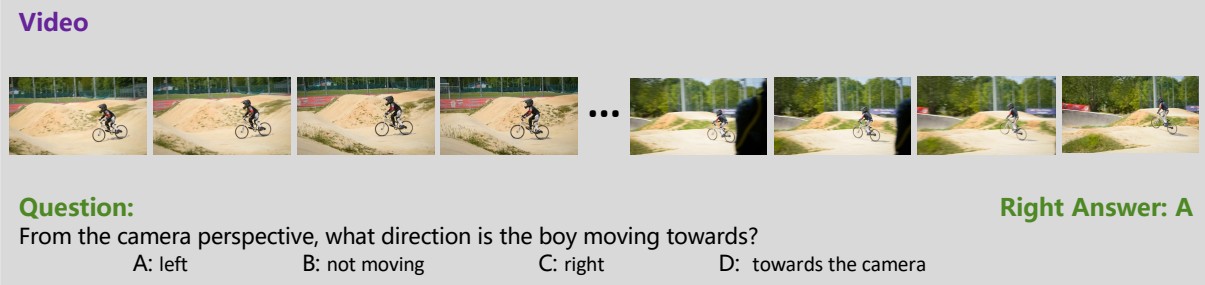

**Question:**                                                                                                    **Right Answer: A**
From the camera perspective, what direction is the boy moving towards?
          A: left                    B: not moving                    C: right                    D: towards the camera

**MLLM-4D:** "**<thinking>**The objective is to determine the direction the boy is moving towards, from the camera's perspective, based on the visual evidence in the video frames. In the initial frame, we observe a boy on a bicycle, positioned towards the right side of the frame, actively navigating a dirt track. The background features a fence, a red banner, and a soccer goal, all appearing relatively stationary relative to the boy's position. **Spatial State (Frame 0):** - Object Center: [0.6944, 0.1615, 3.9354] - Camera Center: [-0.0243, -0.0013, 0.0003] Analyzing the temporal flow, the visual evidence shows the boy on the bicycle progressively moving further into the scene from the right. As the sequence progresses, the boy's figure appears to grow larger in the frame, and the background elements, such as the red banner and the soccer goal, shift from the right side of the frame towards the left, indicating that the camera is also moving. The dirt mounds in the foreground also appear to flow past the camera, reinforcing the sense of forward motion.** By the final frame, the boy is significantly closer to the camera, and the background elements have shifted considerably to the left, confirming that the boy is moving towards the camera's initial position.**Spatial State (Frame 4):** - Object Center: [0.8192, 0.2294, 5.4275] - Camera Center: [-0.0249, -0.0011, 0.0002]** Synthesizing the motion trajectory: the boy on the bicycle consistently moves from the right side of the frame towards the center and then further into the scene, appearing to approach the camera's original viewpoint. The camera itself also exhibits a slight leftward pan, but the primary visual cue of the boy's increasing size and the background elements receding to the left strongly indicate that the boy is moving directly towards the camera. This visual progression directly supports the choice of "towards the camera" as the most accurate estimation.**</thinking>** **<answer>A</answer>**" ✅

**VG-LLM:** "The video sequence shows the boy moving from the left side of the frame towards the right. The camera angle remains constant throughout the video, capturing the boy's movement in a consistent direction. **Based on this observation, the boy is moving towards the right. Therefore, the final answer is: C** " ❌

**Qwen3-VL:**" Step 1: Observe the video frames. In the initial frames (0:00 to 0:01), the boy is seen riding **towards the right side of the frame.** Step 2: Continue observing. At 0:02, the boy is still moving to the right, but now the camera is panning to follow him, and the boy is moving away from the camera's initial position. Step 3: At 0:03, **the boy is still moving to the right**, and the camera is now following him from behind, indicating that he is moving away from the camera's initial position.Step 4: **At 0:04, the boy is still moving to the right**, and the camera is still following him from behind. **Therefore, the final answer is: C**" ❌

*Figure 16.* Qualitative comparison on VLM4D benchmark.

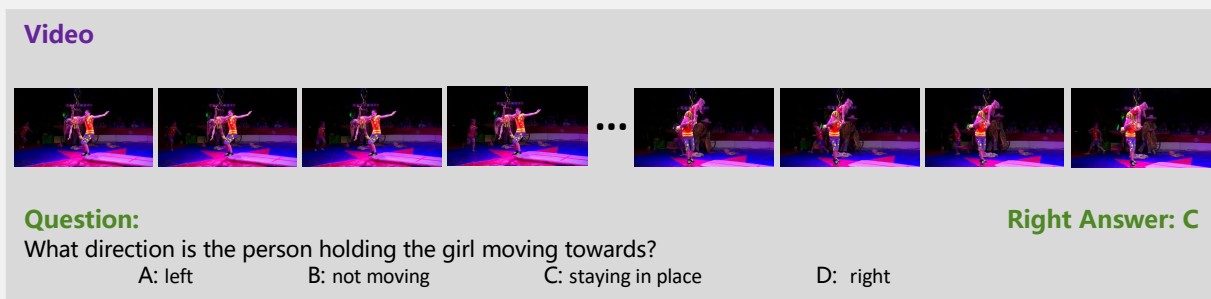

**Video**

**Question:**                                               **Right Answer: C**

What direction is the person holding the girl moving towards?

       A: left           B: not moving           C: staying in place         D: right

**MLLM-4D:** "**\<thinking\>**The objective is to determine the direction of movement of the person holding the girl during the sequence. In the initial frame, we observe the person holding the girl positioned towards the left side of the performance area. The girl is balanced on the person's shoulders, and their combined figure is relatively static against the background of the circus tent. **\*\*Spatial State (Frame 0.000):\*\* - Object Center: [-0.2448, 0.2527, 3.5453] - Camera Center: [-0.0293, -0.0022, 0.0011] Analyzing the temporal flow, the visual evidence shows that the person holding the girl maintains a remarkably consistent position throughout the sequence. There is no discernible lateral shift of their overall body mass across the stage. Their feet remain planted in a similar spot on the blue and red mat, and their upper body orientation relative to the stage and the background elements, such as the distant audience and the circus props, does not change significantly. While the girl performs a slight adjustment in her posture, the base person holding her exhibits no noticeable translation or rotation that would indicate movement in any cardinal direction.** By the final frame, the visual evidence confirms that the person holding the girl is still in a nearly identical spatial location as observed at the beginning. The background elements appear static, and the person's silhouette occupies the same general area of the stage. **\*\*Spatial State (Frame 2.999):\*\* - Object Center: [-0.2426, 0.2452, 3.5385] - Camera Center: [-0.0293, -0.0022, 0.0011]** Synthesizing the motion trajectory: the person holding the girl demonstrates a lack of significant translational movement across the stage. Their position remains largely fixed, with no observable drift to the left, right, or forward/backward. This visual progression directly supports the choice of "staying in place" as the most accurate estimation.**\</thinking\>\<answer\>C\</answer\>**" ✅

**VG-LLM:** "**The person holding the girl is moving towards the right side of the frame. The girl is being held above their head and then lowered down.** This movement is consistent throughout the video. **Therefore, the final answer is: D** " ❌

**Qwen3-VL:**" Step 1: Observe the person holding the girl in the video frames. In the initial frames, the person holding the girl is standing still, holding her up. Step 2: As the video progresses, the person holding the girl begins to move. **Specifically, they start to move to the right side of the frame**, as seen from 0:02 to 0:03. Step 3: The movement is clear and consistent towards the right side of the stage. Step 4: Therefore, the person holding the girl is moving towards the right. **Therefore, the final answer is: D**" ❌

*Figure 17.* Qualitative comparison on VLM4D benchmark.

