# OpenReview forum: "MLLM-4D: Towards Visual-based Spatial-Temporal Intelligence"
_ICML.cc/2026/Conference — ICML 2026 regular_

### Official Review · Reviewer_jRdc · 2026-03-05

**Soundness:** 4
**Presentation:** 4
**Significance:** 4
**Originality:** 4
**Overall Recommendation:** 4
**Confidence:** 3

**Summary:**

This paper proposes MLLM-4D, a framework for improving multimodal large language models in 4D spatiotemporal reasoning (3D + time) for dynamic videos. It introduces an automated data engine that converts stereo video datasets into millions of 4D instruction–answer pairs, and trains the model with a two-stage post-training pipeline incorporating Spatiotemporal Chain-of-Thought (ST-CoT) and a physics-aware Spatiotemporal Reward (ST-Reward) to encourage physically consistent reasoning and reduce hallucinations.

**Compliance With Llm Reviewing Policy:**

Affirmed.

**Final Justification:**

After reading the rebuttal, I believe the authors have addressed the major concerns reasonably well. Therefore, I maintain my original rating.

**Key Questions For Authors:**

- Does MLLM-4D also improve performance on standard static 3D spatial reasoning benchmarks, such as VSI-Bench[1], ScanQA[2], or SQA3D[3]?
- Have the authors considered integrating monocular depth estimation and camera pose estimation models to broaden the applicability of the data engine?
- The current experiments restrict inputs to 32 frames. How sensitive is the model to the frame sampling strategy, and are there plans to extend the framework to longer video sequences?


[1] Thinking in Space: How Multimodal Large Language Models See, Remember and Recall Spaces, CVPR 2025

[2] ScanQA: 3D Question Answering for Spatial Scene Understanding, CVPR 2022

[3] SQA3D: Situated Question Answering in 3D Scenes, ICLR 2023

**Limitations:**

yes

**Strengths And Weaknesses:**

Strengths:
- Constructs a large-scale high-quality 4D instruction dataset and a evaluation benchmark.
- The proposed ST-CoT and ST-Reward are well-motivated and technically interesting.
- The paper is well organized, and experiments are clear and convincing.

Weaknesses:
- Experiments mainly focus on VLM4D and the proposed MLLM4D-Bench. Results on widely used static spatial reasoning benchmarks are missing.
- The data engine heavily relies on stereo video datasets for accurate point clouds and camera poses, which may limit the diversity of training data.

---

> ### Author Rebuttal · Authors · 2026-03-31
>
> We sincerely thank the reviewer for the insightful feedback. We appreciate for the comments that our work constructs **“large scale high quality” 4D QA dataset,** proposes **“well-motivated and technically interesting” ST-CoT and ST-Reward, and** shows **“well organized”** paper and **“clear and convincing”** experiments**.** We hope our response can address your remaining concerns.
>
> ----
>
> **W1&K1: More Evaluation**
>
> It is important to note that many existing baselines (e.g., VLM-3R, Spatial-MLLM) are directly supervised on data tailored specifically for benchmarks like VSI-Bench, ScanQA, or SQA3D. We further conduct evaluation on recent proposed benchmarks such as MMSI-Bench, ERQA, and OSI-Bench, which are suggested by other reviewer. **The results can be found in the table in W1 part of our response to Reviewer Wv6p, which demonstrate the generalization and spatiotemporal reasoning capabilities of MLLM4D over the baselines.**
>
> ----
>
> **W2&K2: broadening applicability to monocular video**
>
> To ensure the robustness of our design, we have already developed and evaluated an alternative **monocular-based data engine** to compare its efficacy against our stereo-based pipeline. First, we employ Gemini2.5-flash to identify all moving entities, followed by Grounded-SAM2 for temporally consistent instance segmentation and tracking. Entity descriptions are further refined using PixelRefer for fine-grained attribute grounding. We sample pixels in each region and apply a 4D tracking method, such as SpatialTrakerV2 to track points in 4D space. Since 4D tracking method typically produces depth at a relative-scale, we incorporate a metricscale depth estimation method, such as MoGe-2 to align the final per-frame object-level points.
>
> We curated a parallel dataset of 2M QA pairs from the same 100k scenes using this monocular pipeline. As shown the table below, while **both large-scale 4D datasets drive significant performance gains over the baseline**, the stereo-derived data consistently yields superior results. These results empirically demonstrate that while our engine *is* applicable to monocular video, the **stereo-based pipeline provides significantly higher-quality supervision**. **Monocular methods, despite their broad availability, still suffer from fundamental depth ambiguity and accumulated spatial drift.**
>
> |  | MLLM4D-Bench | VLM4D |
> | --- | --- | --- |
> | SFT with 2M QA from monocular videos | 59.9 | 56.2 |
> | SFT with MLLM4D-2M | **70.1** | **59.7** |
>
> ----
>
> **K3: Frame sensitivity and future work**
>
> While 32 frames is the standard for spatial or  spatiotemporal reasoning in related works(e.g., VG-LLM, Spatial-MLLM, and VLM-3R), we further provide a sensitivity analysis by reducing frames to 16 and 8 on VLM4D. **As shown in the table below, our MLLM-4D and baseline methods are robust to frame sampling**. We agree that extending 4D reasoning to longer videos is a compelling direction (see line 760-767), and we believe MLLM-4D can serve as foundation for future integration with long-context architectures.
>
> |  | 32 |  16 | 8 |
> | --- | --- | --- | --- |
> | Qwen3-VL | 52.2 | 54.3 | 53.3 |
> | VLM-3R | 33.9 | 34.5 | 34.5 |
> | VG-LLM | 46.5 | 47.1 | 46.7 |
> | Our MLLM-4D | **61.0** |  **61.3** | **61.0** |

---

> > ### Author Rebuttal · Reviewer_jRdc · 2026-04-01
> >
> > Thank you to the authors for their reply. I am happy with the provided answers and have no further questions at this time. I will update my score pending the reviewer discussions that follow.

---

> > > ### Author Response · Authors · 2026-04-04
> > >
> > > Thank you for your thoughtful reviews and positive recommendation. We are glad that our responses have addressed your concerns. The updated results and clarifications will be included in the revised paper.

---

### Official Review · Reviewer_9gGi · 2026-03-10

**Soundness:** 2
**Presentation:** 3
**Significance:** 3
**Originality:** 2
**Overall Recommendation:** 4
**Confidence:** 4

**Summary:**

This paper introduces MLLM-4D, a comprehensive framework designed to enhance spatiotemporal understanding and reasoning. The authors develop a cost-efficient data curation pipeline that annotates existing stereo video datasets with 4D metadata to construct high-quality instructional data. This effort yields the MLLM4D-2M dataset for Supervised Fine-Tuning (SFT), a 7K CoT cold-start dataset, and the MLLM4D-R1-30k dataset for Reinforcement Fine-Tuning (RFT), alongside the MLLM4D-Bench for comprehensive evaluation. Additionally, the authors propose a three-stage post-training strategy, comprising SFT, CoT-based SFT, and RFT. Extensive experiments demonstrate that the proposed MLLM-4D achieves state-of-the-art performance in 4D understanding and reasoning tasks.

**Compliance With Llm Reviewing Policy:**

Affirmed.

**Final Justification:**

My concerns have been fully addressed. I recommend that the authors incorporate these experiments into the revised manuscript, and I am happy to raise my rating accordingly.

**Key Questions For Authors:**

Please refer to the weaknesses part.

**Limitations:**

yes

**Strengths And Weaknesses:**

Strengths:

1.	This work develops a scalable pipeline to generate 4D metadata from stereo videos, contributing significant resources (2M SFT data, 30k RFT data, and a new benchmark) to the field.
2.	The three-stage training strategy is well-motivated for progressively enhancing model capabilities. Furthermore, the design of the Spatiotemporal Reward effectively aligns the model's internal reasoning with 4D dynamics, achieving this without requiring modifications to the underlying model architecture.
3.	The method achieves state-of-the-art results on 4D reasoning tasks, demonstrating significant improvements over current MLLMs.

Weaknesses:

1.	The automated pipeline for metadata extraction lacks quantitative validation. This raises concerns about noise propagation into the ground-truth labels used for training the MLLM4D-2M dataset.
2.	The ST-CoT framework compels the model to generate intermediate reasoning variables—such as spatiotemporal objectives, temporal boundaries, spatial states, and geometric deltas. However, the accuracy of these intermediate steps remains unvalidated. Furthermore, the ablation studies are insufficient as they fail to compare ST-CoT against a standard Chain-of-Thought (CoT) baseline.
3.	The evaluation is primarily confined to the proposed MLLM4D-Bench and the VLM4D benchmark. However, VLM4D predominantly emphasizes translational and rotational motion queries—categories that are also covered in the MLLM4D-2M training data.
4.	The paper lacks assessment on established general spatiotemporal understanding benchmarks (e.g., VSI-Bench, MMSI-Bench, ERQA), which would better demonstrate the model's out-of-distribution generalization capability

---

> ### Author Rebuttal · Authors · 2026-03-31
>
> We sincerely thank the reviewer for the insightful feedback. We appreciate for the recognition that our work contributes **“significant resources” to the field,** employs **“well-motivated” training strategy, and** demonstrates **“significant improvement”.** We hope our response can address your remaining concerns.
>
> ----
>
> **W1: Robustness of data engine and quantitative validation.**
>
> Please refer to **W1 part** of our response to **Reviewer 9hWH for** robustness of data engine**.** For quantitative validation**, as shown in the** **Data Scaling Curve (Figure 5), our model’s performance improves consistently as the data volume increases.  Our positive scaling trend confirms that our filtering protocols ensure our data engine is robust to noise.**
>
> ----
>
> **W2: ST-CoT Validation**
>
> To ensure the accuracy of intermediate generated variables (e.g., geometric deltas, spatial states) of ST-CoT, we uniquely designed the ST-Reward function, which provides explicit reinforcement signals to align these intermediate 4D outputs with ground-truth physics and geometry, effectively acting as a continuous validation mechanism during training.
>
> To address the ablation concerns, **we conduct comparison between standard CoT (general step-by-step reasoning) and our ST-CoT**. The results below show that while standard CoT improves general logic, our ST-CoT specifically captures the intricate spatiotemporal dependencies required for 4D tasks. When ST-CoT combined with ST-Reward, it achieves a state of verifiable reasoning where the intermediate variables are forced to align with 4D states.
>
> | VLM4D Benchmark | MLLM-4D-SFT + standard CoT | MLLM-4D-SFT + ST-CoT | MLLM-4D-RFT + standard CoT + standard GRPO | MLLM-4D-RFT + ST-CoT + ST-reward GRPO |
> | --- | --- | --- | --- | --- |
> | methods | 59.7 | 60.0 | 61.0 | 63.1 |
>
> ----
>
> **W3&W4: More evaluation about generalization capabilities of MLLM4D**
>
> As our MLLM-4D is designed to achieve visual spatiotemporal intelligence that focus on enhancing 4D reasoning from video, we prioritize comparison on 4D spatiotemporal benchmarks such as MLLM4D-Bench and VLM4D. The concurrent work DSR even limit their comparison with baseline only on self-constructed benchmark.
>
> To further verify the generalization capabilities, as some baselines (such as VLM-3R and Spaital-MLLM) directly construct data suit for VSI-Bench, we conduct more comparison on other recent proposed benchmarks, such as the reviewer wanted MMSI-Bench, ERQA, and OSI-Bench (**table in W1 part of our response to Reviewer Wv6p**). **The results better demonstrate out of distribution generalization capability of our MLLM-4D.**

---

> > ### Author Rebuttal · Reviewer_9gGi · 2026-04-03
> >
> > My concerns have been fully addressed. I recommend that the authors incorporate these experiments into the revised manuscript, and I am willing to raise my rating accordingly.

---

> > > ### Author Response · Authors · 2026-04-04
> > >
> > > Thank you for your insightful reviews and positive recommendation. We are glad that our responses have addressed your concerns, and will incorporate the experimental results into the revised manuscript as suggested.

---

### Official Review · Reviewer_Wv6p · 2026-03-11

**Soundness:** 2
**Presentation:** 2
**Significance:** 2
**Originality:** 2
**Overall Recommendation:** 4
**Confidence:** 4

**Summary:**

This paper introduces MLLM-4D, including (1) a benchmark, (2) training datasets for SFT and RL, and (3) a family of MLLMs powerful at 4D spatiotemporal reasoning. The authors carefully designed automatic pipelines for data generation. Experiments under the self-conducted MLLM4D-Bench and VLM4D demonstrate the effectiveness of the proposed data, training recipes, and spatiotemporal reward.

**Compliance With Llm Reviewing Policy:**

Affirmed.

**Final Justification:**

After checking the rebuttal, all my concerns are addressed.

**Key Questions For Authors:**

Please check the "weaknesses" section.

**Limitations:**

Yes.

**Strengths And Weaknesses:**

**Strengths**
1. Overall, this paper is not hard to follow.
2. The motivation is clear and reasonable.
3. The data pipeline is carefully designed.


**Weaknesses**
1. The generalization capabilities of MLLM-4D are unclear. It is encouraged to evaluate on related vision-centric general benchmarks, including both image and video benchmarks.
2. Missing human performance. As illustrated in Figures 1, 11, 12, and 13, I found it difficult for humans to answer those questions precisely without extra camera meta information. Therefore, human performance of MLLM4D-Bench should be included. Moreover, in-depth analysis of when humans are better than models and vice versa is also encouraged.

---

> ### Author Rebuttal · Authors · 2026-03-31
>
> We sincerely thank the reviewer for the valuable feedback and comments regarding our work **“clear and reasonable motivation” and “carefully designed” data pipeline.** We hope our response can address your remaining concerns.
>
> ----
>
> **W1: Generalization on vision-centric general benchmarks**
>
> We clarify that MLLM-4D is designed to achieve visual spatiotemporal intelligence that focus on enhancing 4D reasoning from pure video ****rather than improve general capabilities. Related work such as VG-LLM(NeurIPS’25), Spatial-MLLM(NeurIPS’25) and VLM-3R(CVPR’26) only conduct evaluation on 3D spatial intelligent benchmark rather than general vision benchmark. It is worth noting that this field is in its early stages, there are relatively few benchmarks specifically evaluating the 4D spatiotemporal reasoning capabilities of MLLMs. The concurrent work DSR[1] even limit their comparison with baseline only on self-constructed benchmark.
>
> To address the concern on generalization, we provide more evaluation on recent proposed spatial intelligence benchmarks ****such as MMSI-Bench[2](image-based QA, we use Motion camera and Motion object subset), OSI-Bench[3](video-based QA, we use Dynamic Metric subset), ERQA[4](embodied reasoning QA, we use Trajectory Reasoning and Action Reasoning subset) and vision-centric general benchmark Video-MME[5]. The results show that our MLLM-4D consistent improvements across heterogeneous tasks and external benchmarks confirm that our model learns fundamental, transferable representations rather than merely overfitting to pipeline-specific heuristics.
>
> |  | MMSI-Bench |  OSIbench | ERQA | Video-MME |
> | --- | --- | --- | --- | --- |
> | Qwen2.5-VL-7B | 27.2 | 15.4 | 40.6 | 55.4 |
> | Qwen3-VL-8B | 27.3 | 27.7 | 38.4 | 58.5 |
> | VLM-3R | 24.0 | 22.8 | 30.4 | 53.3 |
> | Spatial-MLLM | 16.0 |  13.2 | 36.2 | 44.9 |
> | VG-LLM | 20.1 | 11.1 | 39.3 | 59.5 |
> | Our MLLM-4D | **31.0** | **30.2** | **42.1** | **62.4** |
>
> [1] Learning to Reason in 4D: Dynamic Spatial Understanding for Vision Language Models. CVPR’26
>
> [2]Mmsi-bench: A benchmark for multi-image spatial intelligence. ICLR’26
>
> [3]From indoor to open world: Revealing the spatial reasoning gap in mllms. CVPR’26
>
> [4]Gemini Robotics brings AI into the physical world.
>
> [5]Video-mme: The first-ever comprehensive evaluation benchmark of multi-modal llms in video analysis. CVPR’25
>
> ----
>
> **W2: Human Performance**
>
> We agree that a human baseline is essential for situating the difficulty of MLLM4D-Bench. To address this, we conducted an user evaluation with 50 volunteers (graduate students with computer vision backgrounds) who independently answered the questions from the MLLM4D-Bench. Despite the lack of explicit camera metadata, humans achieved an average accuracy of 87.3% (Abs.Dis 88.2%, Rel.Dir. 86.5%, Abs.Dis. 87.8%, Abs.Dis. 86.9%, Rel.Dis. 87.1%, and Rel.Dir. 87.3%), outperforming current models in 4D reasoning.
>
> In-depth Analysis: Humans > Models: Humans excel in relational dynamics and Spatiotemporal reasoning, leveraging strong common-sense priors to fill in missing temporal information;
> Models > Humans: Interestingly, MLLM-4D shows superior precision in fine-grained spatial quantification (e.g., cases where motion is subtle and options are numerically close), where human perception is naturally more qualitative and subjective. These results suggest that while MLLM-4D is closing the gap in 4D reasoning, there remains a reasoning ceiling that human-level common sense provides.

---

> > ### Author Rebuttal · Reviewer_Wv6p · 2026-04-04
> >
> > I am not very convinced by the rebuttal, especially for the first question. The authors only selected a subset for EACH general benchmark. The subset performance can be reported and even emphasized, but at least the overall performance should be reported.

---

> > > ### Author Response · Authors · 2026-04-07
> > >
> > > We appreciate the reviewer’s thoughtful comments. We clarify that the results on Video-MME are overall accuracy. In our initial rebuttal above, we focused on the 4D-related subsets of MMSI-Bench, OSI-Bench, and ERQA because the objective of our work is to advance visual spatiotemporal intelligence (e.g., 4D reasoning) of MLLMs. Since these benchmarks are general spatial intelligence benchmark including both 3D and 4D reasoning tasks, we believed that evaluating the 4D-specific subsets provided the most direct and rigorous validation of our motivation.
> > >
> > > We further provide the overall performance comparison in the table below. Our MLLM-4D consistently outperforms the 3D spatial reasoning baselines (VG-LLM, Spaital-MLLM and VLM-3R) and matches or exceeds the general models like Qwen3-VL in overall accuracy. The overall and 4D-specified results confirm that our model learns fundamental, transferable representations rather than merely overfitting to our pipeline-specific heuristics. We hope the comprehensive evaluation and analysis can address your remaining concerns.
> > >
> > > |  | MMSI-Bench |  OSIbench | ERQA | Video-MME |
> > > | --- | --- | --- | --- | --- |
> > > | Qwen2.5-VL-7B | 27.6 | 22.9 | 42.3 | 55.4 |
> > > | Qwen3-VL-8B | **30.7** | 33.6 | 42.0 | 58.5 |
> > > | VLM-3R | 27.9 | 25.3 | 31.8 | 53.3 |
> > > | Spatial-MLLM | 25.8 | 13.3 | 35.8 | 44.9 |
> > > | VG-LLM | 27.5 | 20.3 | 39.3 | 59.5 |
> > > | Our MLLM-4D | 29.6 | **35.6** | **42.5** | **62.4** |

---

### Official Review · Reviewer_9hWH · 2026-03-13

**Soundness:** 4
**Presentation:** 3
**Significance:** 4
**Originality:** 3
**Overall Recommendation:** 5
**Confidence:** 4

**Summary:**

This paper proposes MLLM-4D, a novel and comprehensive framework to enhance the spatial-
temporal intelligence of MLLMs without requiring architectural modifications. An automated data curation pipeline to generate high-quality 4D spatiotemporal instructional data is designed
by repurposing the existing stereoscopic video datasets.
Leveraging this pipeline, this paper proposes the MLLM4D-2M
and MLLM4D-R1-30k datasets for SFT and RFT, along-
side MLLM4D-Bench for comprehensive evaluation. The specialized ST-CoT
prompting strategies and physics-grounded ST-reward are integrated into GRPO for RFT. The experiments demonstrate that Qwen3-VL-8B based MLLM-4D
achieves state-of-the-art 4D understanding and reasoning
performance with only RGB video input on MLLM4D-Bench, with an average score of 72.7%.

**Compliance With Llm Reviewing Policy:**

Affirmed.

**Final Justification:**

My concerns are fully addressed, and I am delighted to keep my recommendation for the paper's acceptance.

**Key Questions For Authors:**

1. How robust is the data engine to errors in camera poses, stereo depth, segmentation, and object tracking?

2. How much of the improvement comes from the scale of synthetic supervision versus the specific ST-CoT and ST-reward design?

3. Can the method generalize to monocular datasets or real-world videos without high-quality stereo metadata at training time?

**Limitations:**

The authors did not discuss the limitations and potential negative societal impact of their work. The main limitation is that its supervision pipeline appears tailored to scenarios where good stereo reconstruction and tracking are available, which may limit coverage of more unconstrained internet-scale videos.

**Strengths And Weaknesses:**

## Strengths

1. The paper is well written and easy to follow. The motivation is clear and problem framing is timely for visual 4D spatiotemporal reasoning from ordinary RGB video.

2. The paper is well engineered with a full-stack design, combining data curation pipeline, large-scale training dataset, benchmarking, and post-training recipe with ST-CoT and ST-reward.

3. Using stereoscopic videos plus estimated camera poses, object-level 3D points, and semantic descriptions to derive QA labels by physics-based computation is a sensible way to scale beyond small manually annotated 4D datasets.

4. It is surprising that a standard MLLM backbone, when paired with better 4D data and post-training, can achieve strong gains, without architecture changes.

5. The post training recipe is well motivated. With ST-CoT and ST-reward, the reported gains are substantial on MLLM4D-Bench and VLM4D. The ablation also suggests SFT is doing most of the heavy lifting, while ST-reward gives an additional improvement.

## Weaknesses

1. The whole pipeline relies heavily on stereo-derived supervision. Error propagation and label noise may occur on stereo video, camera pose estimation, depth, segmentation/tracking, and object description generation.

2. The benchmark may inherit the paper’s own pipeline assumptions. Because the same physical decomposition and solver logic underlie the dataset and benchmark design, there is a risk that the task distribution favors models trained to match those conventions. The VLM4D transfer only partially addresses this concern.

3. The benchmark does not yet cover richer 4D reasoning, such as occlusion persistence, counterfactual motion, or multi-object relational dynamics beyond relatively geometric QA.

---

> ### Author Rebuttal · Authors · 2026-03-31
>
> We sincerely thank the reviewer for the constructive feedback. We appreciate the comments regarding our work is **“clear motivated” to achieve visual 4D spatiotemporal reasoning from pure video, a“full-stack design” with “sensible” data curation pipeline and “well-motivated”post training recipe, and gains “surprisingly strong” performance.** We hope our response can address your remaining concerns.
>
> ----
>
> **W1&K1: Robustness of data engine**
>
> **To mitigate error propagation, we actually implement a rigorous multi-stage filtering pipeline rather than using raw outputs**. Specifically, for stage 1: (1) to obtain high-quality 4D spatial-temporal metadata, we enforce strict geometric constraints, including a 1-pixel vertical flow threshold and stereo cycle-consistency checks. We utilize DeepLabv3 semantic masking to prune drifting tracks on static surfaces and employ SIFT-based temporal analysis to discard artifacts like cross-fades (detail refer to section 3.1 of Stereo4D paper), we will add this details in L159. (2) During segmentation and tracking, we discard instances with discontinuous 2D masks to ensure reliability. We prune objects where the generated descriptions are inconsistent with the observed active descriptions. For stage2: to obtain high-quality QA pairs we use some data filtering and balancing protocols in line 697-707 of the appendix. For stage3: to obtain high-quality reasoning data, please refer to line 255-264.
>
> **As quantitative results shown in the** **Data Scaling Curve (Figure 5), our model’s performance improves consistently as the data volume increases.  Our positive scaling trend confirms that our filtering protocols ensure our data engine is robust to noise.**
>
> ----
>
> **W2: Generalization capabilities of MLLM4D**
>
> As our MLLM-4D is designed to achieve visual spatiotemporal intelligence that focus on enhancing 4D reasoning from video, we prioritize comparison on 4D spatiotemporal benchmarks such as MLLM4D-Bench and VLM4D. It is worth noting that this field is in its early stages, there are relatively few benchmarks specifically evaluating the 4D spatiotemporal reasoning capabilities of MLLMs. The concurrent work DSR[1] even limit their comparison with baseline only on self-constructed benchmark.
>
> To further verify the generalization capabilities, we conduct more comparison on several recent proposed general spatial reasoning benchmarks, such as MMSI-Bench, OSI-Bench and ERQA (**table in W1 part of our response to Reviewer Wv6p**). The results show that our MLLM-4D consistent improvements across heterogeneous tasks and external benchmarks confirm that **our model learns fundamental, transferable representations rather than merely overfitting to pipeline-specific heuristics.**
>
> [1] Learning to Reason in 4D: Dynamic Spatial Understanding for Vision Language Models. CVPR’26
>
> ----
>
> **W3: 4D Reasoning Richness**
>
> We agree that higher-order reasoning about 4D motion are helpful. We contend that **robust dynamic geometric reasoning is the fundamental bottleneck for current general or 3D spatial reasoning MLLMs**. Establishing a model’s ability to understand 3D geometry over time is a prerequisite for more complex physical or higher-order reasoning. Our work has already moves a essential step toward 4D reasoning.
>
> We conduct a specialized evaluation on 1k samples focused on multi-object relational dynamics, which requires the model to reason about  the distance of an object at frame j to another object at frame i. As shown in the table below, **our MLLM-4D significantly outperforms existing baselines on multi-object relational dynamics task**.
>
> |  | Qwen2.5-VL | Qwen3-VL | VLM-3R | VG-LLM | Our MLLM-4D |
> | --- | --- | --- | --- | --- | --- |
> | Multi-Object Abs. Dis | 22.2 | 23.3 | 19.2 | 52.4 | **66.8** |
>
> ----
>
> **K2: Sources of Performance Gain**
>
> The individual contributions are detailed in **Sec. 5.3 (Ablation Studies). Fig. 5** quantifies the impact of data scaling, while **Table 3** demonstrates the orthogonal gains from ST-CoT and ST-Reward independent of scale. These results indicate that while scale provides a robust foundation, our methodological innovations are crucial for structuring the 4D reasoning process. For more detail about ST-CoT, please refer to **W2 part** of our response to **Reviewer 9gGi**
>
> ----
>
> **K3&L1: Generalization to Monocular Videos**
>
> Please refer to **W2&K2 part** of our response to **Reviewer jRdc.**

---

> > ### Author Rebuttal · Reviewer_9hWH · 2026-04-04
> >
> > Thank you to the authors for the reply. My concerns are fully addressed, and I am delighted to keep my recommendation for the paper's acceptance.

---

> > > ### Author Response · Authors · 2026-04-04
> > >
> > > Thank you for your constructive reviews and positive recommendation. We are glad that our responses have addressed your concerns, and will update all the comments in the final version.

---

### Decision · Program_Chairs · 2026-04-30

**Decision:**

Accept (regular)

**Comment:**

This submission introduces MLLM-4D, a framework enhancing the 4D spatiotemporal reasoning of MLLMs via an automated stereo-video data curation pipeline, ST-CoT, and a physics-grounded ST-reward.

The initial reviews were broadly positive regarding the method's full-stack design and empirical gains but raised strict, valid concerns regarding data engine robustness, out-of-distribution generalization, and the absence of a human baseline. During the rebuttal, the authors provided a highly rigorous defense that directly resolved these empirical gaps. They demonstrated the model's generalization across multiple external benchmarks, validated the data engine's robustness via a monocular pipeline ablation, and executed a 50-person user study to establish a human performance ceiling.

Overall, this paper provides solid technical contributions that the community can build upon, so I would like to recommend it for acceptance.